# ADAPT: Attentive Self-Distillation and Dual-Decoder Prediction Fusion for Continual Panoptic Segmentation

**Ze Yang, Shichao Dong, Ruibo Li, Nan Song & Guosheng Lin** *
College of Computing and Data Science
Nanyang Technological University
50 Nanyang Ave, Singapore 639798
{ze001,shichao001,ruibo001,nan001}@e.ntu.edu.sg, gslin@ntu.edu.sg

## Abstract

Panoptic segmentation, which unifies semantic and instance segmentation into a single task, has witnessed considerable success on predefined tasks. However, traditional methods tend to struggle with catastrophic forgetting and poor generalization when learning from a continuous stream of new tasks. While continual learning aims to mitigate these challenges, our study reveals that existing continual panoptic segmentation (CPS) methods often suffer from efficiency or scalability issues. To address these limitations, we propose an *efficient adaptation* framework that incorporates *attentive self-distillation* and *dual-decoder prediction fusion* to efficiently preserve prior knowledge while facilitating model generalization. Specifically, we freeze the majority of model weights, enabling a shared forward pass between the teacher and student models during distillation. Attentive self-distillation then adaptively distills useful knowledge from the old classes without being distracted from non-object regions, which effectively enhances knowledge retention. Additionally, query-level fusion (QLF) is devised to seamlessly integrate the output of the dual decoders without incurring scale inconsistency. Our method achieves state-of-the-art performance on ADE20K and COCO benchmarks. Code is available at `https://github.com/Ze-Yang/ADAPT`.

## 1 Introduction

Panoptic segmentation Kirillov et al. (2019b;a); Xiong et al. (2019); Wang et al. (2020); Cheng et al. (2020); Strudel et al. (2021); Cheng et al. (2021; 2022) is a fundamental computer vision task that provides a comprehensive understanding of visual scenes Li et al. (2024); Dong et al. (2024) by unifying semantic and instance segmentation. It predicts semantic masks for *stuff* classes—amorphous background regions without distinct object boundaries, such as sky, road, and grass—and instance masks for *thing* classes, which are countable objects with well-defined boundaries like people, cars, and animals. This holistic task is crucial for various applications, including autonomous driving Cordts et al. (2016), robotics Ros et al. (2015), and image editing Aksoy et al. (2018).

In real-world scenarios, models are often required to continually adapt to new classes and evolving data distributions Yang et al. (2020; 2022). To this end, continual learning (CL) has been developed to equip models with the capability to learn from a sequence of tasks while maintaining previously acquired knowledge. A key challenge in CL is known as *catastrophic forgetting*, where the model gradually loses information learned from previous tasks during new training stages. This issue has been extensively studied in classification Li & Hoiem (2017); Rebuffi et al. (2017), object detection Shmelkov et al. (2017); Liu et al. (2023) and semantic segmentation Cermelli et al. (2020); Douillard et al. (2021); Yang et al. (2023).

Despite this progress, continual panoptic segmentation (CPS) remains an under-investigated area with existing approaches exhibiting notable limitations. Concretely, CoMFormer Cermelli et al. (2023) finetunes the entire model on new tasks while employing knowledge distillation (KD) to

---

*Guosheng Lin is the corresponding author.

mitigate forgetting. However, this approach presents three potential risks. First, finetuning the entire model grants excessive plasticity, increasing the risk of *overfitting*. Second, although the final outputs are constrained by the distillation loss, the substantial plasticity in the intermediate layers can still lead to *catastrophic forgetting*. Lastly, updating the entire model using KD necessitates separate forward passes for the teacher and student models, significantly raising *computational costs*. Conversely, ECLIPSE Kim et al. (2024) seeks to preserve base performance by freezing most model weights. This strategy inevitably constrains the model's capacity to generalize to new tasks due to *restricted plasticity*. Moreover, ECLIPSE continuously introduces additional learnable query features and embeddings for each new task, which may cause *scalability issues* as the number of tasks grows.

In this paper, we propose a novel approach for CPS that effectively balances base knowledge retention and learning new tasks. Building on Mask2Former Cheng et al. (2022), we freeze the image encoder and pixel decoder to preserve base knowledge and enhance efficiency. Notably, this weight-freezing design allows for a shared forward pass between the teacher and student models, thereby significantly reducing computational overhead. By selectively finetuning only the cross-attention layers and feed-forward networks in the transformer decoder, our **adaptation strategy** optimally balances between plasticity (learning new information) and rigidity (retaining old knowledge). In addition, the computational overhead of our approach remains nearly constant, regardless of the number of continual learning steps.

To further alleviate forgetting, we develop an **attentive self-distillation** mechanism, inspired by focal loss Lin et al. (2017). Existing KD methods Cermelli et al. (2020); Douillard et al. (2021); Yang et al. (2023) often treat all entities (pixels or instances) uniformly, leading to an overemphasis on dominant background regions that provide little benefit in preserving prior knowledge. In contrast, our approach adaptively re-weights the contribution of each entity in the distillation loss based on the background confidence predicted by the teacher model. This modulated distillation can effectively concentrate on learning informative entities, enhancing the retention of previous knowledge.

Finally, recognizing that KD can still accumulate errors over successive learning steps, we propose a **dual-decoder prediction fusion** paradigm, where we retain the transformer decoder trained on the base dataset for subsequent inference. Specifically, we utilize this fixed decoder to predict base classes while the adapted decoder predicts novel classes learned up to the current step. To combine the predictions from these two decoders, we devise a robust query-level fusion (QLF) strategy to avoid the scale inconsistency issue that occurs in another naive solution — probability-level fusion (PLF). Extensive experiments on ADE20k and COCO benchmarks showcase that our method outperforms state-of-the-art approaches in CPS.

In summary, our approach, **ADAPT**, offers an efficient and scalable solution to CPS. Our contributions can be summarized as follows:

- We present an *efficient adaptation* strategy that freezes the image encoder and pixel decoder, enabling a shared forward pass between the teacher and student models during distillation, significantly reducing computational overhead.

- We develop an *attentive self-distillation* loss, which emphasizes informative entities (or queries) and down-weights less useful ones based on background confidence, improving the preservation of prior knowledge.

- We devise a *dual-decoder prediction fusion* mechanism, dubbed as query-level fusion (QLF), which combines the outputs of the base and adapted decoders without relying on probability fusion, effectively preventing the scale inconsistency issue present in PLF.

- Our method outperforms current CPS approaches, achieving superior results on ADE20k and COCO benchmarks.

## 2 RELATED WORKS

### 2.1 PANOPTIC SEGMENTATION

Panoptic segmentation, introduced by Kirillov et al. (2019b), aims to unify semantic and instance segmentation, with Panoptic FPN Kirillov et al. (2019a) extending this concept using a feature pyra-

mid network for both *stuff* and *thing* classes. UPSNet Xiong et al. (2019) improved efficiency by introducing a learnable panoptic head, while Axial-DeepLab Wang et al. (2020) leveraged axial attention to enhance spatial feature representation. Panoptic-DeepLab Cheng et al. (2020) streamlined panoptic segmentation using pixel-level classification in a fully convolutional framework. Transformers further advanced the field with MaskFormer Cheng et al. (2021), which introduced mask classification, simplifying segmentation. Mask2Former Cheng et al. (2022) extended this by providing a generalized, transformer-based framework for universal segmentation tasks, achieving state-of-the-art performance. For a fair comparison, we adopt the Mask2Former framework, consistent with prior works on continual panoptic segmentation Cermelli et al. (2023); Kim et al. (2024).

## 2.2 Continual Panoptic Segmentation

Continual panoptic segmentation remains a relatively under-explored field, with initial efforts focusing on addressing the challenge of catastrophic forgetting. Building upon the transformer-based architecture Mask2Former Cheng et al. (2022), CoMFormer Cermelli et al. (2023) employs knowledge distillation and mask-based pseudo-labeling to mitigate forgetting when finetuning the entire model. However, this approach yields suboptimal base class performance due to error accumulation over continual learning steps. ECLIPSE Kim et al. (2024) improves base class retention by freezing the majority of model weights and progressively introduces learnable prompt features to handle new tasks. While this strategy improves rigidity, it limits the model's plasticity and raises scalability issues as additional prompts are required for each task. To resolve these limitations, we propose a novel approach that effectively preserves base knowledge while maintaining strong generalization ability, without incurring scalability issues.

## 2.3 Self Distillation

Unlike traditional knowledge distillation Hinton et al. (2015), which typically employs larger teacher models to guide smaller student models, self-distillation leverages the same architecture to transfer knowledge gained in earlier stages to improve performance in later stages. This technique has been applied across various domains to mitigate forgetting, including classification Li & Hoiem (2017); Rebuffi et al. (2017); Zhang et al. (2021), object detection Shmelkov et al. (2017); Liu et al. (2023), semantic segmentation Cermelli et al. (2020); Douillard et al. (2021); Yang et al. (2023), panoptic segmentation Cermelli et al. (2023), and image generation Song et al. (2024). However, most of these methods require separate forward passes for the teacher and student models, resulting in considerable computational overhead. To overcome this limitation, we propose to share a single forward pass for the majority of modules between the teacher and student models. By freezing the model up to the pixel decoder and adapting only the final decoder, our approach achieves a double benefit—improving efficiency and retaining base knowledge.

## 3 Method

### 3.1 Preliminary

**Problem Setting.** Panoptic segmentation is a challenging task that aims to unify semantic and instance segmentation. It predicts semantic masks for 'stuff' classes (amorphous, background regions without distinct object boundaries, e.g., sky, road, grass) and instance masks for 'thing' classes (countable object classes with distinct boundaries, e.g., people, cars, animals). In the context of continual learning, the objective is to train a model across a sequence of tasks (or steps) $t = 0, 1, 2, \ldots, T$. At each step $t$, a new training set $\mathcal{D}^t = \{(x_j, y_j)\}_{j=1}^{N_t}$ is available, where each input image $x_j$ contains at least one new-class mask label, i.e., $|y_j| \neq 0$. Importantly, at step $t$, only the current class labels $y_j \in \mathcal{C}^t$ are available, while the labels from previous steps $\mathcal{C}^{0:t-1}$ or future steps $\mathcal{C}^{t+1:T}$ are inaccessible, even though these classes may still appear in the input image. This scenario is known as the *overlapped* setting Cermelli et al. (2020). Following the convention in continual learning, the class subsets across tasks are disjoint. A special class label $\emptyset$ represents "no object" detected for a given query, akin to the "background" class in traditional segmentation methods Chen et al. (2017). Once all learning steps are completed, the model is evaluated on the full set of classes $\mathcal{C}^{0:T}$ encountered throughout training.

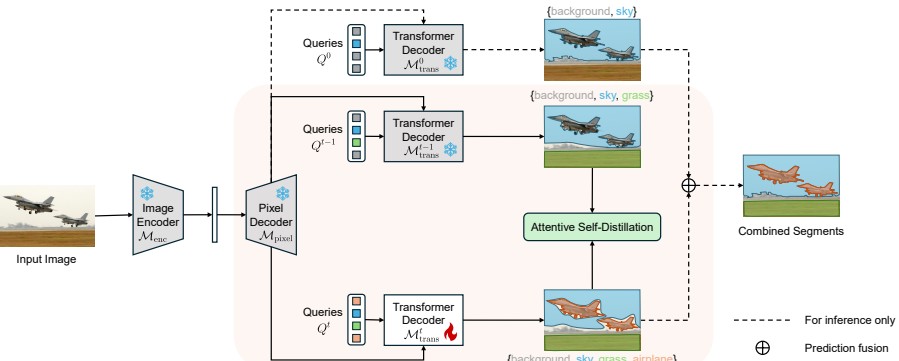

Figure 1: Overview of the proposed dual-decoder framework for continual panoptic segmentation. During training, attentive self-distillation is applied between $\mathcal{M}_{\text{trans}}^{t-1}$ and $\mathcal{M}_{\text{trans}}^{t}$. During inference, the fixed base decoder $\mathcal{M}_{\text{trans}}^{0}$ predicts base classes, while the adapted decoder $\mathcal{M}_{\text{trans}}^{t}$ handles novel classes. We apply query-level fusion (QLF) to effectively combine the outputs of these two decoders and preventing scale mismatches. More details can be found in Sec. 3.2 and Sec. 3.3.

**Model Architecture.** We adopt Mask2Former Cheng et al. (2022), a widely used transformer-based model for universal segmentation tasks. Given an input image $x$, the image encoder $\mathcal{M}_{\text{enc}}$ first extracts low-resolution features. These features are then progressively upsampled by the pixel decoder $\mathcal{M}_{\text{pixel}}$ to generate high-resolution per-pixel embeddings. The transformer decoder $\mathcal{M}_{\text{trans}}$ utilizes a set of learnable object queries $Q = \{q_i\}_{i=1}^{N_q}$, which interact with the feature pyramid output by the pixel decoder via cross-attention. For each query $q_i$, the decoder outputs a binary mask $M_i \in \mathbb{R}^{\frac{H}{s} \times \frac{W}{s}}$ with associated class probabilities $p_x^t(i, c) \in \mathbb{R}^{N_c+1}$, where $H$ and $W$ denote the spatial dimensions of the input image, $s$ is the stride and $N_c$ the number of classes. This architecture offers two key advantages: (1) compared to per-pixel classification Long et al. (2015), mask classification enables better differentiation between objects, making it ideal for panoptic segmentation, and (2) its DETR-style Carion et al. (2020) end-to-end set prediction eliminates the need for a cumbersome two-stage process He et al. (2017), resulting in a more efficient and streamlined framework.

## 3.2 ADAPTATION STRATEGY

Continual panoptic segmentation has been scarcely explored. The existing method, CoM-Former Cermelli et al. (2023), finetunes the entire model on the new sequence of tasks, which often leads to significant degradation in the performance of base classes. To mitigate this issue, Kim et al. Kim et al. (2024) freeze most of the model's weights, allowing only the newly introduced learnable queries (or prompts) to be updated. While this mechanism helps maintain base class performance (rigidity), it compromises the model's ability to generalize to new tasks (plasticity). In this work, we freeze the image encoder $\mathcal{M}_{\text{enc}}$ and pixel decoder $\mathcal{M}_{\text{pixel}}$ for efficiency and systematically investigate the effects of fine-tuning different components of the transformer decoder $\mathcal{M}_{\text{trans}}$, which consists of $L$ transformer blocks each with self-attention layers $\{\text{SA}_i\}_{i=1}^{L}$, cross-attention layers $\{\text{CA}_i\}_{i=1}^{L}$ and feed-forward networks $\{\text{FFN}_i\}_{i=1}^{L}$. We observe that freezing the image encoder $\mathcal{M}_{\text{enc}}$ and pixel decoder $\mathcal{M}_{\text{pixel}}$ significantly reduces catastrophic forgetting while preserving considerable generalization capacity. Additionally, it enables the feasibility of our self-distillation design, which relies on the shared and frozen weights. For the transformer decoder $\mathcal{M}_{\text{trans}}^{t}$, we fine-tune only the cross-attention layers and the feed-forward networks (FFN), striking an ideal balance between rigidity and plasticity, along with high efficiency. We initialize the learnable queries $Q^t$ from $Q^{t-1}$ and follow Cermelli et al. (2023) for classifier initialization.

## 3.3 ATTENTIVE SELF-DISTILLATION

Even though only the transformer decoder is updated during continual learning, the forgetting issue persists as the previous class labels $\mathcal{C}^{0:t-1}$ are no longer available. To mitigate this issue, we resort to knowledge distillation, which penalizes the discrepancies between the predicted class probabilities of the teacher model $\mathcal{T}$ and student model $\mathcal{S}$ given the same input. Following the prior method Yang et al. (2023) in continual semantic segmentation, we opt to establish correct class correspondences for knowledge distillation by modifying the old model outputs. Notably, the natural bias toward

newly learned classes in continual learning results in false negatives (FN) for base classes and false positives (FP) for novel classes, degrading both base and novel class performance. Consequently, the key objective is to mitigate this bias using the knowledge from the previous classes $\mathcal{C}^{0:t-1}$. To this end, we apply knowledge distillation only to the queries $\mathcal{B}$ that are not matched with any new classes $\mathcal{C}^t$ in the bipartite matching Carion et al. (2020), which can be formulated as:

$$\ell_{\text{kd}}^{\theta^t}(x,y) = -\frac{1}{|\mathcal{B}|} \sum_{i \in \mathcal{B}} \sum_{c \in \mathcal{C}^{0:t} \cup \{\emptyset\}} \hat{p}_x^{t-1}(i,c) \log p_x^t(i,c), \tag{1}$$

where $p_x^t(i,c)$ refers to the probability of class $c$ for query $q_i^t$ predicted by $\mathcal{M}_{\text{trans}}^t$, and $\hat{p}_x^{t-1}(i,c)$ is the probability $p_x^{t-1}(i,c)$ predicted by $\mathcal{M}_{\text{trans}}^{t-1}$, expanded with zero probability for the added novel classes $c \in \mathcal{C}^t$, formally as:

$$\hat{p}_x^{t-1}(i,c) = \begin{cases} 0 & \text{if } c \in \mathcal{C}^t \\ p_x^{t-1}(i,c) & \text{otherwise}. \end{cases} \tag{2}$$

However, this distillation loss treats all queries equally, despite the fact that over 95% of queries contain `no object`. This can cause the model to overemphasize these dominant `no object` regions, which contribute little to preserving previously learned knowledge. To address this issue, we propose adaptively re-weighting the contribution of each query within the distillation loss. Inspired by focal loss Lin et al. (2017), we incorporate a modulating term $\alpha(1 - \hat{p}_x^{t-1}(i,\emptyset))^\gamma$ into the distillation loss, resulting in our adaptive weighted version:

$$\bar{\ell}_{\text{kd}}^{\theta^t}(x,y) = -\frac{1}{|\mathcal{B}|} \sum_{i \in \mathcal{B}} \alpha(1 - \hat{p}_x^{t-1}(i,\emptyset))^\gamma \sum_{c \in \mathcal{C}^{0:t} \cup \{\emptyset\}} \hat{p}_x^{t-1}(i,c) \log p_x^t(i,c), \tag{3}$$

where $\gamma \geq 0$ is a tunable focusing parameter and $\alpha > 0$ is a scaling factor. When $\gamma > 1$, this modulating term down-weights queries that are predicted as `no object` with high confidence ($\hat{p}_x^{t-1}(i,\emptyset) \to 1$) by the old model. The parameter $\gamma$ controls the shape of the re-weighting curve. Specifically, the larger value of $\gamma$ extends the flat range of the curve and down-weights more `no object` queries with lower confidence. In addition, the scaling factor $\alpha$ allows the modulating term to be greater than one, making it possible to emphasize the contribution of `object` queries ($\hat{p}_x^{t-1}(i,\emptyset) \to 0$). When $\gamma = 0$, the proposed adaptive re-weighted KD loss reduces to the standard non-re-weighting version. We empirically observe that $\gamma = 3$ and $\alpha = 4$ yields the best results.

**Discussion.** We highlight the distinction of our adaptive re-weighted self-distillation in three aspects. First, thanks to our model freezing and adaptive decoder design, the teacher model and student model share the same weights up to the pixel decoder $\mathcal{M}_{pixel}$. Unlike traditional knowledge distillation (KD) based methods Cermelli et al. (2023); Douillard et al. (2021), which require separate forward passes for the teacher model and student model, our approach performs only a single forward pass except for the final dual decoders $\mathcal{M}_{\text{trans}}^{t-1}$ and $\mathcal{M}_{\text{trans}}^t$, manifesting higher efficiency. Second, in contrast to Kim et al. (2024), which encounters scalability issues due to the continual introduction of new prompt features and embeddings for each new task, our design remains invariant to the number of tasks $T$ or the number of classes $|\mathcal{C}^t|$ within each task $t$, allowing for greater scalability. Third, distinct to Cermelli et al. (2023), which applies unbiased KD Cermelli et al. (2020) to all queries, our approach selectively applies modulated KD only to queries that are not matched with any new class, thereby maintaining old knowledge more effectively and efficiently.

## 3.4 DUAL-DECODER PREDICTION FUSION

Although knowledge distillation has been adopted to alleviate catastrophic forgetting, it tends to accumulate errors over successive continual learning steps, devastating old knowledge learned in the early stage. This challenge motivates us to retain the transformer decoder $\mathcal{M}_{\text{trans}}^0$ trained on the base dataset $\mathcal{D}^0$, as it contains precise knowledge for decoding useful information from the feature pyramid output by the pixel decoder $\mathcal{M}_{pixel}$. In particular, only the weights of the cross-attention layers and the feed-forward networks need to be stored, since the self-attention layers are fixed and shared across all learning steps. Notably, the retained base decoder does not participate in the training phase, resulting in no additional computational overhead. During inference, the base decoder $\mathcal{M}_{\text{trans}}^0$ is used to predict masks for the base classes $\mathcal{C}^0$, while the adapted decoder $\mathcal{M}_{\text{trans}}^t$ handles the new classes $\mathcal{C}^{1:t}$ learned up to the current step $t$. Following Cheng et al. (2022); Cermelli et al. (2023), we use the Softmax activation function to calculate the final class scores.

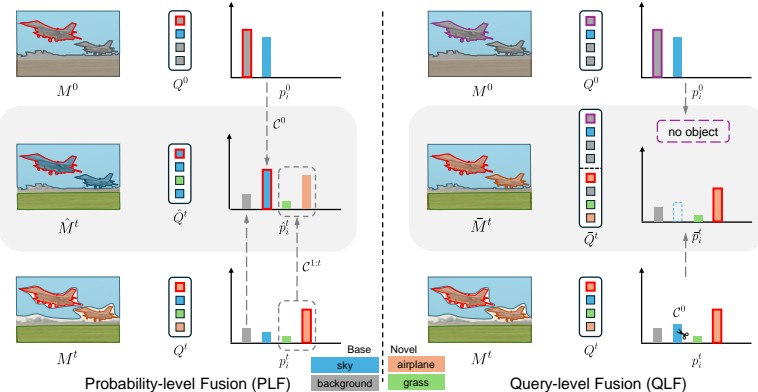

Figure 2: Illustration of Probability-Level Fusion (PLF) and Query-Level Fusion (QLF). Each query predicts a segment with associated class scores. The query square and corresponding segment edge are bolded in the same color (red or purple) to indicate their correspondence.

**Probability-level fusion.** As the transformer decoders $\mathcal{M}^0_{\text{trans}}$ and $\mathcal{M}^t_{\text{trans}}$ are responsible for predicting base and novel classes respectively, an effective mechanism is needed to combine their outputs for the final predictions. One straightforward method, which we refer to as probability-level fusion (PLF), is to replace the base-class $c \in \mathcal{C}^0$ probability scores $p^t_{i,c}$ for each query $i$ predicted by $\mathcal{M}^t_{\text{trans}}$ with the corresponding scores $p^0_{i,c}$ predicted by $\mathcal{M}^0_{\text{trans}}$. For brevity, here we omit $x$ in $p^t_x(i, c)$ and use the subscript version $p^t_{i,c}$. After obtaining the combined class scores for each query, we need to determine their associated binary masks. Specifically, the corresponding mask $\hat{M}^t_i$ for the $i$-$th$ query $\hat{q}^t_i$ is derived by selecting between the masks $M^0_i$ and $M^t_i$, which are generated by the base decoder $\mathcal{M}^0_{\text{trans}}$ and the adapted decoder $\mathcal{M}^t_{\text{trans}}$, respectively. The selection process is determined by the combined probability score $\hat{p}^t_{i,c}$ and is defined as follows:

$$\hat{p}^t_{i,c} = \begin{cases} p^0_{i,c} & \text{if } c \in \mathcal{C}^0, \\ p^t_{i,c} & \text{otherwise,} \end{cases} \qquad \hat{M}^t_i = \begin{cases} M^0_i & \text{if } \hat{c}_i \in \mathcal{C}^0, \\ M^t_i & \text{otherwise.} \end{cases} \tag{4}$$

Here, $\hat{c}_i = \arg\max_c \hat{p}^t_{i,c}$ denotes the class with the highest combined probability score. This mechanism ensures that masks predicted by the base decoder are used for base classes ($c \in \mathcal{C}^0$), while masks from the adapted decoder are utilized for novel classes ($c \in \mathcal{C}^{1:t}$). However, this approach can lead to inconsistencies between the probability scales of base and novel classes, as they are not optimized on the same data distribution. We observe that base-class probabilities tend to have larger values than novel-class probabilities, which can lead to confusion between base and novel classes. For instance, as shown in Fig. 2, the novel class "airplane" is misclassified as the base class "sky" due to this discrepancy in probability-level fusion.

**Query-level fusion.** In light of the inconsistency issue in PLF, we propose query-level fusion (QLF) as a simple yet effective alternative. As illustrated in Fig. 2, we discard the base-class probabilities $p^t_{i,c} (c \in \mathcal{C}^0)$ predicted by $\mathcal{M}^t_{\text{trans}}$ so that the adapted decoder only makes predictions for novel classes. Conversely, $\mathcal{M}^0_{\text{trans}}$ is solely responsible for predicting the base classes. In this way, the "airplane", though with high confidence in the base class "sky", can be correctly suppressed by the "no object" class with even higher confidence. Otherwise, the "airplane" will be misclassified as "sky", as occurred in PLF (See left side of Fig. 2). During query-level fusion, the more accurate sky mask with a finer boundary, predicted by the base decoder, supersedes the less precise version generated by the adapted decoder, successfully eliminating error accumulation. This process is achieved using the standard panoptic inference procedure (see Algorithm 1 in Appendix) in Mask2Former Cheng et al. (2022).

## 4 EXPERIMENTS

### 4.1 DATASET AND EVALUATION METRICS

We validate the effectiveness of our approach on ADE20K Zhou et al. (2017) and COCO Lin et al. (2014) benchmarks. ADE20K consists of 25,574 training images and 2,000 validation images. This

dataset includes a total of 150 classes, divided into 100 "thing" classes and 50 "stuff" classes. Unlike COCO Lin et al. (2014), with an average of 7.7 instances and 3.5 classes per image, and Pascal VOC Everingham et al. (2010), with 2.3 instances and 1.4 classes per image, ADE20K averages 19.5 instances and 9.9 classes per image. The complexity of scenes in ADE20K makes it a challenging benchmark for continual panoptic segmentation. COCO consists of 118,287 training images and 5,000 validation images with 133 classes. We leave the COCO results in the Appendix. We adopt the conventional Panoptic Quality (PQ) Kirillov et al. (2019b) metric used in panoptic segmentation, which is defined as the product of Recognition Quality (RQ) and Segmentation Quality (SQ):

$$PQ = RQ \times SQ = \underbrace{\frac{|TP|}{|TP| + \frac{1}{2}|FP| + \frac{1}{2}|FN|}}_{\text{Recognition Quality (RQ)}} \times \underbrace{\frac{\sum_{(p,g)\in TP} \text{IoU}(p,g)}{|TP|}}_{\text{Segmentation Quality (SQ)}},$$

where $\text{IoU}(p,g)$ is the Intersection-over-Union between a predicted segment $p$ and a ground truth segment $g$. $TP$, $FP$, and $FN$ represent true positives, false positives, and false negatives, respectively. RQ measures how well the model detects and classifies objects, while SQ evaluates how well the predicted segments match the ground truth in terms of shape and overlap.

## 4.2 BASELINES AND CONTINUAL PROTOCOLS

We adhere to the continual panoptic segmentation protocol from ECLIPSE Kim et al. (2024), evaluating both $100-n$ ($n = 5, 10, 50$) and $50-n$ ($n = 10, 20, 50$) scenarios. For instance, $100-10$ refers to base training with 100 classes, followed by 5 incremental steps, each introducing 10 new classes. Note that a smaller $n$ results in more continual steps, making the scenario more challenging. We re-implement the representative methods MiB Cermelli et al. (2020) and PLOP Douillard et al. (2021), originally developed for continual semantic segmentation, as competitive baselines. Additionally, we include the state-of-the-art methods CoMFormer Cermelli et al. (2023) and ECLIPSE Kim et al. (2024), which are specifically designed for continual panoptic segmentation, in our evaluation. Alongside these, we also evaluate the simplest baseline, FT (Fine-Tuning), which finetunes the base model on new tasks without any specific continual learning design. All experiments are conducted under the *overlap* setting, following the latest state-of-the-art approach Kim et al. (2024). After the final step $T$, we evaluate performance on base classes ($\mathcal{C}^0$), novel classes ($\mathcal{C}^{1:T}$), and the full set of classes ($\mathcal{C}^{0:T}$, denoted as $all$).

## 4.3 IMPLEMENTATION DETAILS

Following state-of-the-art methods Cermelli et al. (2023); Kim et al. (2024), we adopt the Mask2Former Cheng et al. (2022) model with an output stride of 4, using ResNet-50 He et al. (2016) as the backbone, unless otherwise stated. To ensure fairness, we follow the same training hyperparameters as our competitors Cermelli et al. (2023); Kim et al. (2024), with the exception of using a higher learning rate, which leads to faster convergence and slightly improved performance. For all settings, we report Panoptic Quality (PQ) results on the standard validation set. Experiments are conducted using two NVIDIA RTX 6000 Ada GPUs on ADE20K and four on COCO. For more details, please refer to A.5.

## 4.4 PERFORMANCE

As shown in Table 1, the FT baseline fails to retain previous knowledge due to the absence of anti-forgetting mechanisms. This leads to catastrophic forgetting, not only of base class knowledge but also of the new classes learned in earlier steps, resulting in poor overall performance on novel classes. MiB Cermelli et al. (2020) and PLOP Douillard et al. (2021) demonstrate significant improvements over FT, thanks to unbiased knowledge distillation and multi-scale local POD distillation. CoMFormer Cermelli et al. (2023) improves performance further, particularly in the $100$-$n$ settings, surpassing PLOP by +5.5 PQ on base classes in the $100-10$ scenario. ECLIPSE Kim et al. (2024) mitigates the forgetting issue by only updating newly introduced prompts, which brings improvements on base classes, such as a +5.2 PQ gain over CoMFormer in the $100-10$ setting. Our approach, FDAS, consistently surpasses all state-of-the-art methods across all scenarios. In the $100-10$ setting, FDAS outperforms CoMFormer and ECLIPSE on both base (+6.2/+1.0 PQ) and novel (+9.2/+9.8 PQ) classes, achieving 42.2 PQ on base classes and 26.3 PQ on novel classes. In

| Method | 100-5 (11 tasks) | | | 100-10 (6 tasks) | | | 100-50 (2 tasks) | | |
|---|---|---|---|---|---|---|---|---|---|
| | *1-100* | *101-150* | *all* | *1-100* | *101-150* | *all* | *1-100* | *101-150* | *all* |
| FT† | 0.0 | 1.3 | 0.4 | 0.0 | 2.9 | 1.0 | 0.0 | 25.8 | 8.6 |
| MiB† Cermelli et al. (2020) | 24.0 | 6.5 | 18.1 | 27.1 | 10.0 | 21.4 | 35.1 | 19.3 | 29.8 |
| PLOP† Douillard et al. (2021) | 28.1 | 15.7 | 24.0 | 30.5 | 17.5 | 26.1 | 41.0 | 26.6 | 36.2 |
| CoMFormer† Cermelli et al. (2023) | 34.4 | 15.9 | 28.2 | 36.0 | 17.1 | 29.7 | 41.1 | 27.7 | 36.7 |
| ECLIPSE‡ Kim et al. (2024) | 41.1 | 16.6 | 32.9 | 41.4 | 18.8 | 33.9 | 41.7 | 23.5 | 35.6 |
| ECLIPSE* Kim et al. (2024) | 39.5 | 14.7 | 31.2 | 41.2 | 16.5 | 33.0 | 39.9 | 21.1 | 33.6 |
| ADAPT (Ours) | **42.3** | **19.8** | **34.8** | **42.2** | **26.3** | **36.9** | **42.5** | **27.8** | **37.6** |
| joint | 43.4 | 32.9 | 39.9 | 43.4 | 32.9 | 39.9 | 43.4 | 32.9 | 39.9 |

(a)

| Method | 50-10 (11 tasks) | | | 50-20 (6 tasks) | | | 50-50 (3 tasks) | | |
|---|---|---|---|---|---|---|---|---|---|
| | *1-50* | *51-150* | *all* | *1-50* | *51-150* | *all* | *1-50* | *51-150* | *all* |
| FT‡ | 0.0 | 1.7 | 1.1 | 0.0 | 4.4 | 2.9 | 0.0 | 12.0 | 8.1 |
| MiB‡ Cermelli et al. (2020) | 34.9 | 7.7 | 16.8 | 38.8 | 10.9 | 20.2 | 42.4 | 15.5 | 24.4 |
| PLOP‡ Douillard et al. (2021) | 39.9 | 15.0 | 23.3 | 43.9 | 16.2 | 25.4 | 45.8 | 18.7 | 27.7 |
| CoMFormer‡ Cermelli et al. (2023) | 38.5 | 15.6 | 23.2 | 42.7 | 17.2 | 25.7 | 45.0 | 19.3 | 27.9 |
| ECLIPSE‡ Kim et al. (2024) | 45.9 | 17.3 | 26.8 | 46.4 | 19.6 | 28.6 | 46.0 | 20.7 | 29.2 |
| ECLIPSE* Kim et al. (2024) | 46.4 | 16.8 | 26.7 | 47.1 | 17.7 | 27.5 | 46.0 | 18.6 | 27.7 |
| ADAPT (Ours) | **49.5** | **21.3** | **30.7** | **49.7** | **26.2** | **34.0** | **49.8** | **28.7** | **35.7** |
| joint | 50.5 | 34.6 | 39.9 | 50.5 | 34.6 | 39.9 | 50.5 | 34.6 | 39.9 |

(b)

Table 1: **Continual panoptic segmentation results (PQ)** on the **ADE20K** Zhou et al. (2017) benchmark, with the number of base classes $|\mathcal{C}^0|$ set to (a) 100 and (b) 50. All methods are based upon the Mask2Former framework Cheng et al. (2022). The *joint* setting indicates that all classes are trained simultaneously in an offline manner. † and ‡ indicate results are taken from Cermelli et al. (2023) and Kim et al. (2024), respectively. * denotes results reproduced using the official code.

the more challenging 50-10 scenario, FDAS exceeds CoMFormer and ECLIPSE by +11.0/+5.7 PQ on base classes and +3.1/+4.5 PQ on novel classes, respectively. Beyond its strong performance, FDAS offers a computational advantage by utilizing a single-forward self-distillation design, unlike MiB, PLOP, and CoMFormer, which require double forward passes. Compared to ECLIPSE, our FDAS balances strong base performance with enhanced generalization capacity, made possible by the complementary fixed and adaptive decoders combined with self-distillation.

## 4.5 ABLATION STUDY

Unless otherwise stated, we conducted ablation studies on the multi-step ADE20K `100-10` setting.

### 4.5.1 OVERALL STUDY

Building upon the baseline CoMFormer Cermelli et al. (2023), Table 2 highlights the complementary contributions of each proposed component, demonstrating their collective impact on the overall performance improvements. Our adaptation strategy improves both base and novel performance, showing enhanced plasticity and rigidity in balance. Incorporating attentive self-distillation further boosts novel class performance since the inherent bias toward new classes is alleviated. As expected, prediction fusion substantially enhances knowledge retention, bridging up the base performance gap with *joint training*. Finally, combining all three components yields the best results, demonstrating that the proposed components are complementary to each other. In summary, our approach offers a well-balanced and efficient solution for continual panoptic segmentation.

We fixed the weights of the image encoder and pixel decoder for two key reasons. First, finetuning these two components does not enhance the model's generalization capacity, yet it substantially increases the risk of overfitting and computational cost, making it an inefficient trade-off. In contrast, freezing the image encoder and pixel decoder helps alleviate the forgetting issue since their weights are preserved to retain base knowledge. Interestingly, comparing the first two rows of Table 2, the transformer decoder itself (ablated in the following paragraph) demonstrates sufficient learning capability to generalize to new tasks. Second, this design allows a large portion of weights to be shared between the teacher model and the student model during knowledge distillation, which we refer to as self-distillation.

| Adaptation Strategy | Attentive Self-Distillation | Predition Fusion | 100-10 (6 tasks) | | |
|:---:|:---:|:---:|:---:|:---:|:---:|
| | | | *1-100* | *101-150* | *all* |
| | | | 36.0 | 17.1 | 29.7 |
| ✓ | | | 38.3 | 20.1 | 32.3 |
| ✓ | ✓ | | 37.2 | 22.6 | 32.4 |
| ✓ | | ✓ | **42.4** | 22.8 | 35.8 |
| ✓ | ✓ | ✓ | 42.2 | **26.3** | **36.9** |

Table 2: **Effect of the proposed components.** Without attentive self-distillation, unbiased KD serves as the baseline. When the adaptation strategy is omitted, weight-shared self-distillation becomes infeasible, and the entire old model is used as the teacher model.

| FFN | Self-att (SA) | Cross-att (CA) | Trainable Params | 100-10 (6 tasks) | | |
|:---:|:---:|:---:|:---:|:---:|:---:|:---:|
| | | | | *1-100* | *101-150* | *all* |
| | | | 0.29M | 42.0 | 5.3 | 29.7 |
| ✓ | | | 9.75M | 42.3 | 23.6 | 36.1 |
| | ✓ | | 2.66M | 42.5 | 16.6 | 33.9 |
| | | ✓ | 2.66M | **42.6** | 17.2 | 34.1 |
| ✓ | ✓ | | 12.12M | 42.1 | 22.5 | 35.6 |
| ✓ | | ✓ | 12.12M | 42.2 | **26.3** | **36.9** |
| | ✓ | ✓ | 5.03M | 42.3 | 18.0 | 34.2 |
| ✓ | ✓ | ✓ | 14.50M | 42.4 | 24.3 | 36.4 |

Table 3: **Impact of fine-tuning different components of the transformer decoder.** FFN, Self-att (SA), and Cross-att (CA) refer to the feed-forward network, self-attention layers, and cross-attention layers, respectively.

### 4.5.2 ADAPTATION STRATEGY

Based on this, we investigate the impact of fine-tuning different components within the transformer decoder, as presented in Table 3. Finetuning the FFN alone yields strong results (23.6 PQ) on novel classes, while cross-attention (CA) or self-attention (SA) alone shows limited generalization ability. It highlights the critical role of FFN in enhancing model generalization. The best performance is achieved when both FFN and CA are finetuned together, reaching 26.3 PQ on novel classes and 36.9 PQ overall. In contrast, finetuning all components or combining FFN with SA does not lead to substantial gains, indicating that the combination of FFN and CA offers the most effective improvement in generalization without overfitting.

### 4.5.3 ATTENTIVE SELF-DISTILLATION

As shown in Table 4, the effect of the attentive self-distillation is influenced by the modulation factors $\alpha$ and $\gamma$. When $\gamma = 0$, the attentive self-distillation loss degrades to the uniform version, which results in poorer performance on novel classes (e.g., PQ drops to 17.1 and 15.7 for $\alpha = 2$ and $\alpha = 4$, respectively) regardless of different values of $\alpha$. This can be attributed to the overwhelming influence of the dominant `no object` regions, which diverts the distillation process away from informative old class objects. As $\gamma$ increases, the attentive self-distillation can effectively focus on informative regions (old classes $\mathcal{C}^{0:t-1}$), resulting in improved novel performance. This is because, without retention of old knowledge, the base classes will be misclassified as novel classes due to the inherent bias in continual learning, leading to false positive predictions for novel classes and hence lowering PQ. In contrast, these mistakes can be effectively corrected by our attentive self-distillation with a concentration on informative old classes. Specifically, the combination of $\alpha = 4$ and $\gamma = 3$ yields the highest overall PQ of 36.9, striking a balance between retaining base knowledge (42.2 PQ) and learning novel classes (26.3 PQ). This demonstrates the effectiveness of re-weighting the contribution of queries based on `no object` confidence, allowing the model to concentrate on preserving previous knowledge while efficiently learning new information.

### 4.5.4 DUAL-DECODER PREDICTION FUSION

As shown in Table 5, without any fusion strategy, the model struggles to retain base knowledge due to accumulated errors over multiple learning steps. Introducing probability-level fusion (PLF), where base-class probabilities are replaced by those from the base decoder, significantly boosts base performance to 43.1 PQ. However, PLF negatively affects novel classes, reducing PQ from

| $\alpha$ | $\gamma$ | **100-10** (6 tasks) | | | $\alpha$ | $\gamma$ | **100-10** (6 tasks) | | |
|---|---|---|---|---|---|---|---|---|---|
| | | *1-100* | *101-150* | *all* | | | *1-100* | *101-150* | *all* |
| 1 | 0 | 42.6 | 17.5 | 34.2 | 1 | 2 | 41.9 | 24.3 | 36.0 |
| 2 | 0 | 42.6 | 17.1 | 34.1 | 2 | 2 | 42.1 | 23.9 | 36.0 |
| 4 | 0 | 42.6 | 15.7 | 33.6 | 4 | 2 | 42.4 | 25.0 | 36.6 |
| 10 | 0 | 42.7 | 10.1 | 31.9 | 10 | 2 | 42.5 | 22.8 | 36.0 |
| 1 | 1 | 42.2 | 23.9 | 36.1 | 1 | 3 | 42.0 | 22.2 | 35.4 |
| 2 | 1 | 42.3 | 23.8 | 36.2 | 2 | 3 | 42.1 | 23.5 | 35.9 |
| 4 | 1 | 42.5 | 22.4 | 35.8 | 4 | 3 | 42.2 | **26.3** | **36.9** |
| 10 | 1 | 42.6 | 20.1 | 35.1 | 10 | 3 | **42.5** | 23.3 | 36.1 |

Table 4: Effect of $\alpha$ and $\gamma$ in the attentive self-distillation loss.

| Fusion Strategy | **100-10** (6 tasks) | | |
|---|---|---|---|
| | *1-100* | *101-150* | *all* |
| Without | 37.2 | 22.6 | 32.4 |
| Probability-level | **43.1** | 20.8 | 35.7 |
| Query-level | 42.2 | **26.3** | **36.9** |

Table 5: Effect of dual-decoder prediction fusion.

| Method | **Training** | | | **Inference** | | | **100-10** (6 tasks) | | |
|---|---|---|---|---|---|---|---|---|---|
| | #Params | #Iters | Time | FLOPs | GPU Mem. | Time | *1-100* | *101-150* | *all* |
| CoMFormer Cermelli et al. (2023) | 44.38M | **4k** | 3.36 hrs | **97.46G** | **4010 MB** | **43.9 ms** | 36.0 | 17.1 | 29.7 |
| ECLIPSE Kim et al. (2024) | **0.55M** | 16k | 3.48 hrs | 99.27G | 5407 MB | 68.7 ms | 41.2 | 16.5 | 33.0 |
| ADAPT (Ours) | 12.12M | **4k** | **2.27 hrs** | 105.76G | 7127 MB | 48.3 ms | **42.2** | **26.3** | **36.9** |

Table 6: **Efficiency comparison on both training and inference phases.** The training time is reported using two NVIDIA RTX 6000 Ada GPUs. FLOPs, GPU memory usage, and time cost are calculated on a per-image, per-device basis during inference with the same device setup.

22.6 to 20.8, due to scale inconsistency between the base and novel class probabilities. In contrast, query-level fusion (QLF) resolves this inconsistency by eliminating the need for probability fusion, resulting in a more balanced performance across both base and novel classes and yielding the highest overall PQ of 36.9.

### 4.5.5 EFFICIENCY

Table 6 highlights the efficiency of our approach. During incremental training, ADAPT requires only 2.27 hours for five training steps, representing a reduction of 34.8% and 32.4% in training time compared to CoMFormer Cermelli et al. (2023) and ECLIPSE Kim et al. (2024), respectively. The reduced training time can be attributed to ADAPT's frozen- and shared-weight design between the teacher and student models, which eliminates the need for dual-model forward passes, a key requirement for knowledge distillation in CoMFormer. Additionally, our efficient adaptation strategy enables a fourfold reduction in the number of training iterations for each step, decreasing from 16k in ECLIPSE Kim et al. (2024) to 4k in our approach. This substantial reduction contributes to the overall decrease in training time. In terms of inference, our method incurs a modest 6.5% increase in FLOPs compared to ECLIPSE. However, this is accompanied by substantial performance gains, particularly in the novel PQ, which improves from 16.5 to 26.3—a remarkable 59.4% increase. Similar improvements are observed when compared to CoMFormer. Furthermore, the inference speed of ADAPT remains comparable to that of CoMFormer, highlighting the efficiency and practicality of our approach. In contrast, ECLIPSE demonstrates slower inference speeds, likely attributable to its unoptimized implementation.

## 5 CONCLUSION

In this paper, we propose a novel approach for continual panoptic segmentation that effectively balances the retention of base knowledge and learning new tasks. Concretely, we present an *efficient adaptation* strategy that freezes the image encoder and pixel decoder, allowing a shared forward pass between the teacher and student models, significantly saving computational costs. Upon this baseline, we introduce an *attentive self-distillation* loss, which emphasizes informative queries and down-weights non-object regions during distillation, to effectively enhance knowledge retention. Additionally, we devise a *query-level fusion* mechanism to combine the predictions from the dual-decoders. It successfully avoids the scale inconsistency issue as occurred in probability-level fusion. Our method, **ADAPT**, demonstrates superior performance on ADE20k and COCO benchmarks, showcasing its effectiveness and scalability in continual learning.

ACKNOWLEDGMENTS

This research is supported by the Agency for Science, Technology and Research (A*STAR) under its MTC Programmatic Funds (Grant No. M23L7b0021). This research is also supported by the MoE AcRF Tier 1 grant (RG14/22).

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

# A  APPENDIX

## A.1  CONTINUAL PANOPTIC SEGMENTATION ON THE COCO BENCHMARK

We evaluate our approach on the COCO panoptic segmentation benchmark Lin et al. (2014), which consists of 118,287 training images and 5,000 validation images distributed across 133 classes. In alignment with Kim et al. (2024), we use 83 base classes and incrementally add 50 additional classes. It is important to note that the class order in COCO panoptic segmentation follows a sequence of "things" and "stuff". We report results based on a randomly shuffled class order as proposed by Kim et al. (2024), given by the following sequence:

```
[1, 3, 10, 47, 58, 9, 88, 16, 126, 120, 17, 129, 35, 119, 59, 57, 54, 90, 75, 38, 80, 48, 131,
56, 95, 25, 43, 2, 68, 110, 32, 14, 29, 11, 7, 52, 83, 102, 84, 73, 5, 45, 117, 93, 87, 46,
118, 34, 61, 19, 77, 111, 63, 98, 130, 66, 79, 97, 33, 86, 127, 104, 64, 49, 36, 6, 91, 50,
112, 8, 65, 132, 92, 27, 122, 22, 51, 85, 115, 28, 89, 70, 62, 12, 101, 108, 125, 123, 39, 81,
20, 40, 41, 114, 128, 74, 18, 99, 100, 60, 30, 124, 69, 37, 13, 23, 116, 55, 26, 121, 71, 67,
106, 133, 42, 107, 105, 109, 82, 103, 76, 94, 24, 15, 78, 53, 21, 96, 72, 113, 44, 31, 4].
```

We compare our method with two baseline approaches, PLOP Douillard et al. (2021) and CoM-Former Cermelli et al. (2023), with the ResNet-50 backbone network under the *overlap* setting. As illustrated in Table 7, our approach outperforms these baselines in terms of *all* class performance.

| Method | **83-5** (11 tasks) | | | **83-10** (6 tasks) | | |
|---|---|---|---|---|---|---|
| | *1-83* | *84-133* | *all* | *1-83* | *84-133* | *all* |
| CoMFormer Cermelli et al. (2023) | 33.5 | 21.9 | 29.1 | 38.3 | **30.7** | 35.5 |
| ECLIPSE Kim et al. (2024) | 44.2 | 18.7 | 34.6 | 44.9 | 21.3 | 36.0 |
| ADAPT (Ours) | **45.3** | **23.2** | **37.0** | **45.7** | 28.8 | **39.3** |
| Joint | 51.1 | 52.1 | 51.5 | 51.1 | 52.1 | 51.5 |

Table 7: Continual panoptic segmentation results (PQ) on the **COCO** Lin et al. (2014) benchmark with the number of base classes set to 83 under the *overlap* setting.

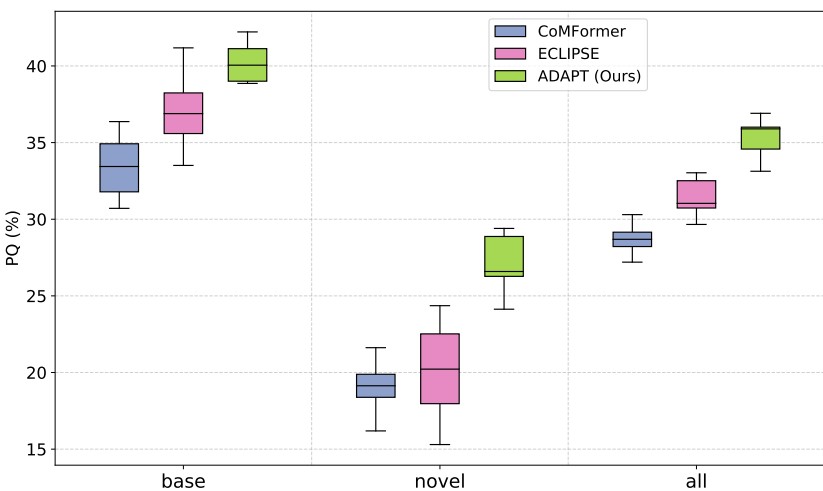

Figure 3: Continual panoptic segmentation result (PQ) distributions for 10 randomly shuffled class orders under the ADE20K Zhou et al. (2017) `100-10` scenario.

## A.2  IMPACT OF CLASS ORDER SHUFFLING ON PERFORMANCE

We examine the robustness of our method with respect to randomly shuffled class orders. We conduct experiments on the ADE20K `100-10` scenario, utilizing the 10 randomly shuffled class orders proposed by Kim et al. (2024) for consistency. The PQ distributions are presented in Figure 3 through boxplots. We note that the results reported in Kim et al. (2024) are affected by an evaluation

| Method | 100-5 (11 tasks) | | | 100-10 (6 tasks) | | | 100-50 (2 tasks) | | |
|---|---|---|---|---|---|---|---|---|---|
| | *1-100* | *101-150* | *all* | *1-100* | *101-150* | *all* | *1-100* | *101-150* | *all* |
| MiB‡ Cermelli et al. (2020) | 20.5 | 4.3 | 15.1 | 27.7 | 7.1 | 20.8 | 33.7 | 10.5 | 26.0 |
| PLOP‡ Douillard et al. (2021) | 19.2 | 8.8 | 15.8 | 28.9 | 10.6 | 22.8 | 34.8 | 12.4 | 27.4 |
| CoMFormer‡ Cermelli et al. (2023) | 20.1 | 8.2 | 16.1 | 29.7 | 10.3 | 23.3 | 34.7 | 13.2 | 27.6 |
| ECLIPSE‡ Kim et al. (2024) | 34.4 | 8.9 | 25.9 | 34.4 | 10.2 | 26.4 | 35.2 | 13.3 | 27.9 |
| ADAPT (Ours) | **36.6** | **9.3** | **27.5** | **36.5** | **11.7** | **28.3** | **36.6** | **18.0** | **30.4** |

Table 8: Continual Panoptic Segmentation results (PQ) on ADE20K Zhou et al. (2017) under the *disjoint* setting. All approaches are based on the same framework Mask2Former Cheng et al. (2022) with the ResNet-50 He et al. (2016) backbone. ‡ indicate results are taken from Kim et al. (2024).

bug, where performance was consistently averaged over the default class order regardless of shuffling. After correcting this issue, we observe that the results deviate from those originally reported in Kim et al. (2024). Notably, our method, ADAPT, exhibits strong resilience to various shuffled class orders, consistently outperforming alternative approaches.

## A.3 CONTINUAL PANOPTIC SEGMENTATION IN THE DISJOINT SETTING

The pioneering work of Cermelli et al. (2020) introduced two distinct settings for continual learning: *disjoint* and *overlap*. Since the *overlap* setting is generally considered more challenging and realistic, we focused primarily on it in our main paper. In this section, we present experimental results for continual panoptic segmentation on ADE20K Zhou et al. (2017) under the *disjoint* setting. The results, as summarized in Table 8, highlight the superior performance of ADAPT over existing methods in continual panoptic segmentation.

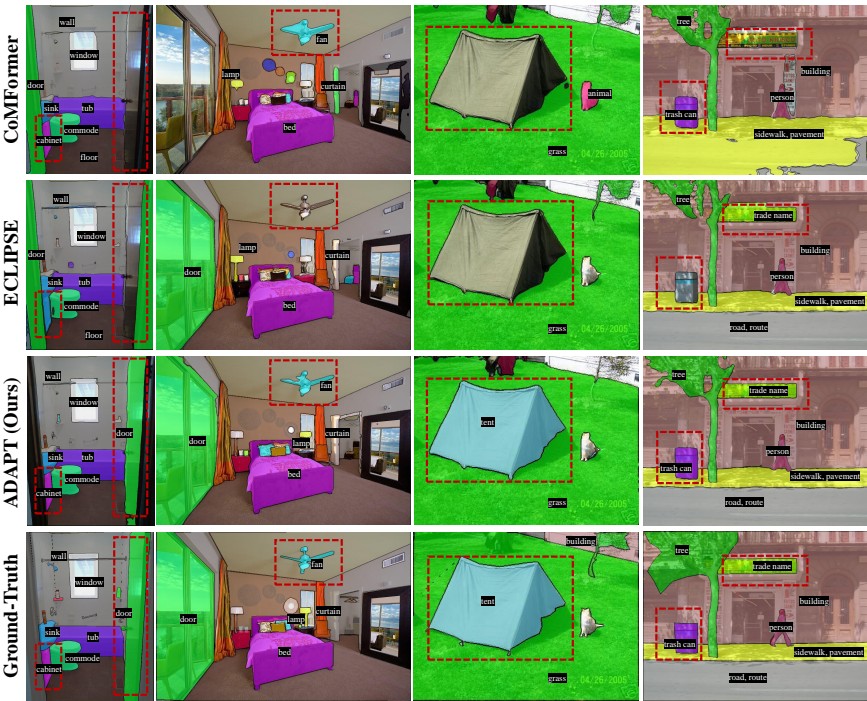

Figure 4: Qualitative visualization for continual panoptic segmentation under the ADE20K Zhou et al. (2017) `100-10` scenario.

### A.4 QUALITATIVE VISUALIZATIONS

We provide qualitative visualizations in Fig. 4 to demonstrate the superior rigidity and plasticity of our approach. Specifically, ADAPT effectively preserves base class knowledge, such as `cabinet` and `door` (first column), while also generalizing well to novel classes, including `fan` (second column), `tent` (third column), `trade name` (last column), and `trash can` (last column).

### A.5 MORE IMPLEMENTATION DETAILS

Following state-of-the-art methods Cermelli et al. (2023); Kim et al. (2024), we adopt the Mask2Former Cheng et al. (2022) model with an output stride of 4, using ResNet-50 He et al. (2016) as the backbone, unless otherwise stated. To ensure fairness, we follow the same training hyperparameters as our competitors Cermelli et al. (2023); Kim et al. (2024), with the exception of using a higher learning rate, which leads to faster convergence and slightly improved performance. The initial learning rate is set to $10^{-4}$ for all steps in the 100-$n$ settings. In the 50-$n$ settings, we use $2 \times 10^{-4}$ for the incremental steps ($t > 0$), while maintaining $10^{-4}$ during base training ($t = 0$). We train the network for 160k iterations during base training, and for 400 iterations per class in all subsequent steps. The batch size is consistently set to 16 across all settings. We use the AdamW optimizer Loshchilov & Hutter (2018) with the same weight decay values as in Cheng et al. (2022). For all settings, we report Panoptic Quality (PQ) results on the standard validation set. Experiments are conducted using two NVIDIA RTX 6000 Ada GPUs on ADE20K and four on COCO.

---

**Algorithm 1** Panoptic Inference with Mask2Former

---

**Require:** $query\_class\_scores$: Tensor of shape [num_queries, num_classes+1]
**Require:** $query\_masks$: Tensor of shape [num_queries, height, width]
**Require:** $num\_classes$: Number of classes
**Require:** $threshold$: Confidence threshold for selecting valid queries
**Require:** $overlap\_threshold$: Minimum overlap threshold for Non-Maximum Suppression (NMS)
**Ensure:** $panoptic\_seg$: Final panoptic segmentation mask
**Ensure:** $segments\_info$: Information of the detected segments
1: Initialize $panoptic\_seg \leftarrow$ zeros($height, width$)
2: Initialize $segments\_info \leftarrow []$
3: Initialize $current\_segment\_id \leftarrow 0$
4:
5: $scores, labels \leftarrow$ max(softmax($query\_class\_scores$), dim=1)
6: $query\_masks \leftarrow$ sigmoid($query\_masks$)
7: $valid \leftarrow (labels \neq background)$ & $(scores > threshold)$
8: $valid\_scores \leftarrow scores[valid]$
9: $valid\_query\_masks \leftarrow query\_masks[valid]$
10:
11: $score\_masks \leftarrow valid\_scores \times valid\_query\_masks$  ▷ [num_valid_queries, height, width]
12: $class\_id\_masks \leftarrow$ argmax($score\_masks$, dim=0)  ▷ [height, width]
13:
14: **for** each query $i$ in valid queries **do**
15:      $pred\_class \leftarrow valid\_scores[i]$
16:      $mask\_area \leftarrow (class\_id\_masks = i).sum()$
17:      $original\_area \leftarrow (valid\_query\_masks[i] > 0.5).sum()$
18:      $mask \leftarrow (class\_id\_masks = i)$ & $(valid\_query\_masks[i] > 0.5)$
19:      **if** $mask\_area > 0$ and $original\_area > 0$ and $mask.sum() > 0$ **then**
20:          **if** $mask\_area/original\_area > overlap\_threshold$ **then**
21:              $current\_segment\_id \leftarrow current\_segment\_id + 1$
22:              $panoptic\_seg[mask] \leftarrow current\_segment\_id$
23:              $segments\_info$.append(
24:                  {
25:                      "id": $current\_segment\_id$,
26:                      "category_id": $pred\_class$
27:                  }
28:              )
29:          **end if**
30:      **end if**
31: **end for**
32: **return** $panoptic\_seg, segments\_info$

---

