# OpenReview forum: "ADAPT: Attentive Self-Distillation and Dual-Decoder Prediction Fusion for Continual Panoptic Segmentation"
_ICLR.cc/2025/Conference — ICLR 2025 Poster_

### Official Review · Reviewer_fZFv · 2024-10-30

**Soundness:** 3
**Presentation:** 3
**Contribution:** 3
**Rating:** 6
**Confidence:** 5

**Summary:**

This paper introduces a novel approach for continual panoptic segmentation, addressing efficiency and scalability challenges faced by existing methods (e.g., CoMFormer’s computational limitations and ECLIPSE’s scalability issues). The authors propose an adaptation strategy that minimizes parameter updates by freezing the majority of model parameters, selectively fine-tuning cross-attention and feedforward layers in the transformer decoder. This is coupled with an attentive self-distillation mechanism to balance plasticity and rigidity effectively. Additionally, a dual-decoder prediction fusion strategy at the query level further enhances model performance. The proposed method, termed ADAPT, is empirically evaluated on the ADE20K dataset, showing improved performance in terms of both plasticity and rigidity over existing approaches.

**Strengths:**

## 1. Strong Motivation and Meaningful Contributions

- The paper is well-motivated, addressing key limitations in current methodologies: (1) CoMFormer’s limited rigidity and computational inefficiency, and (2) ECLIPSE’s restricted plasticity and scalability challenges.
- The proposed solution addresses these limitations effectively. Specifically, the selective fine-tuning of cross-attention and feedforward layers is a meaningful contribution that demonstrates the improved balance between model plasticity and rigidity.


## 2. Impressive Experimental Results

- ADAPT achieves state-of-the-art performance, significantly outperforming existing methods in continual panoptic segmentation tasks.
- The experimental results demonstrate the effectiveness of the proposed solution, with a notable performance margin over alternative approaches.

**Weaknesses:**

# Weaknesses

## 1. Limited Insight into Mechanism Behind Parameter Freezing (L.209)

- While cross-attention layers and feedforward networks are activated, self-attention layers are frozen, as detailed in L.209.
- However, the authors provide only empirical results (Table 2) without an in-depth analysis or hypothesis to support this choice.
- A theoretical justification would strengthen the contribution.

## 2. Lack of Qualitative Analysis

- The paper would benefit from qualitative analysis to complement the quantitative results.
- Such insights would enhance the reader’s understanding and provide explicit visual support for the model’s performance.

## 3. Absence of Comparative Analysis on Training Efficiency

- Given that efficiency is one of the claimed contributions, a comparative analysis on training efficiency, such as training iteration throughput or GPU memory usage, would strengthen the paper.
- A comparison between ADAPT, CoMFormer, and ECLIPSE on these training efficiency metrics would provide a fuller view of ADAPT’s performance.

## 4. Insufficient Details in Figure 2

- Figure 2 lacks clarity, particularly regarding color representations (e.g., which color represents a query or a specific class, such as sky).
- It is also unclear which instances are misclassified in L.315-23.
- Section 3.4 does not clarify (despite having 8 queries in the query-level fusion,) why the probability distribution includes only 4 scores.
- More detailed descriptions of the example base class set, new class set, and the query and color representations would aid comprehension.

## 5. Ambiguity in Baseline Setting (Table 5)

- Table 5 lacks clarity regarding the baseline setting.
- The baseline results without any additional components appear identical to CoMFormer’s results.
- Clarification on whether this is a coincidence or an intentional baseline setting using CoMFormer would help clarify the impact of each proposed component.

## 6. Probability Inconsistency Analysis

- It is unclear which activation function was used to obtain class probabilities.
- Mask2Former and CoMFormer use softmax, whereas ECLIPSE utilizes sigmoid.
- If softmax was used, probability inconsistency may arise due to its relative scoring.
- But, If sigmoid was used, probability inconsistency may reduced.
-  An analysis comparing softmax and sigmoid’s impact on class probability consistency would be beneficial.


# Additional Comments

## A. Experiments in the Disjoint Setting

- While the overlap setting is standard in continual segmentation research, additional results under the disjoint setting would provide further insight and help fellow researchers contextualize the findings.


## B. Expanded Experiments

- Additional evaluations on (1) the COCO panoptic segmentation benchmark, and (2) experiments with shuffled class ordering, as seen in PLOP and ECLIPSE, would offer robustness analysis on ADAPT’s handling of class ordering, enhancing its utility for future research.


## C. Clarification on Reported Performance Metrics

- In Table 1, performance scores for FT, MiB, PLOP, and CoMFormer are cited from ECLIPSE, while ECLIPSE’s score is reproduced.
- This may cause confusion, as it’s unclear why certain scores are excerpted versus reproduced.
- Given ADAPT’s strong performance across metrics, it may be beneficial to rely on previously reported scores for consistency.

**Questions:**

Please check the weaknesses.

## Justification.

Overall, this paper makes significant contributions to the field of continual panoptic segmentation, demonstrating state-of-the-art results in both plasticity and rigidity. However, several areas require additional refinement to meet the high standards expected at ICLR, including more rigorous analysis, qualitative insights, and additional experimental details.

## Recommendation

Although my initial recommendation is 5: marginally below the acceptance threshold, my real rating is the Borderline.
My final decision will depend on the authors' rebuttal and any revisions addressing the outlined weaknesses.

---

> ### Author Response · Authors · 2024-12-01
> **Response to Reviewer fZFv (1/2)**
>
> Thank you for your positive feedback on our motivation, contributions, and performance gain. We sincerely appreciate your constructive comments, which have helped us improve our manuscript. Below, we provide our point-by-point responses to address the concerns raised:
>
> ---
> **Q1:** Insight into Parameter Freezing
>
> **A1:** The **self-attention layers** operate on the encoded feature space, which extracts general semantic information (such as relationships between regions or objects) that remains useful across tasks. Freezing these layers prevents the model from overwriting these learned relationships, helping to generalize to unseen tasks. The **cross-attention layers** align object queries (representing instances or candidates) with the encoded feature map of the image. Differentiating classes often relies on distinct features; for example, the number of legs helps distinguish a person from a dog, while color and texture differentiate a horse from a zebra. Fine-tuning the cross-attention layers enables the model to adapt to novel classes by attending to these distinctive features for each query. Finally, the **feed-forward networks (FFN)** facilitate better feature interaction and generalization by bridging the gap between the frozen self-attention layers and the active cross-attention layers, enabling more task-specific learning while maintaining stability.
>
> ---
> **Q2:** Lack of Qualitative Analysis
>
> **A2:** We provide qualitative visualizations in **Figure 4** in Appendix of the revised manuscript to demonstrate the superior rigidity and plasticity of our approach. Specifically, ADAPT effectively preserves base class knowledge, such as *cabinet* and *door* (first column), while also generalizing well to novel classes, including *fan* (second column), *tent* (third column), *trade name* (last column), and *trash can* (last column).
>
> ---
> **Q3:** Efficiency Analysis
>
> **A3:** The table below (**Table 6** in the revised manuscript) highlights the efficiency of our approach. During incremental training, ADAPT requires only 2.27 hours for five training steps, representing a **reduction of 34.8\% and 32.4\% in training time** compared to CoMFormer and ECLIPSE, respectively. The reduced training time can be attributed to ADAPT's frozen- and shared-weight design between the teacher and student models, which eliminates the need for dual-model forward passes for knowledge distillation in CoMFormer.
> Additionally, our efficient adaptation strategy enables a **fourfold reduction in the number of training iterations for each step**, decreasing **from 16k in ECLIPSE to 4k** in our approach. This substantial reduction contributes to the overall decrease in training time.
> In terms of inference, our method incurs a modest 6.5\% increase in FLOPs compared to ECLIPSE. However, this is accompanied by substantial performance gains, particularly in novel PQ, which improves from **16.5 to 26.3—a remarkable 59.4\% increase**. Similar improvements are observed when compared to CoMFormer. Furthermore, the inference speed of ADAPT remains comparable to that of CoMFormer, highlighting the efficiency and practicality of our approach. In contrast, ECLIPSE demonstrates slower inference speeds, likely attributable to its unoptimized implementation. We have added the efficiency analysis to **Sec. 4.5.5**.
>
> |**Method**||**Training** ||| **Inference**|||**100-10 (6 tasks)** ||
> |:-|:-:|:-:|:-:|:-:|:-:|:-:|:-:|:-:|:-:|
> ||#Params|#Iters|Time|FLOPs|GPU Mem.|Time|*1-100*|*101-150*|*all*|
> |CoMFormer|44.38M|**4k**|3.36 hrs|**97.46G**|**4010 MB**|**43.9 ms**|36.0|17.1|29.7|
> |ECLIPSE|**0.55M**|16k|3.48 hrs|99.27G|5407 MB|68.7 ms|41.2|16.5|33.0|
> |**ADAPT (Ours)**|12.12M|**4k**|**2.27 hrs**|105.76G|7127 MB|48.3 ms|**42.2**|**26.3**|**36.9**|
>
> ---
> **Q4:** Insufficient Details in Figure 2
>
> **A4:** We have added color representations in Figure 2, where "sky" (blue) represents the base class, and "grass" (green) and "airplane" (orange) are the novel classes at time step t. Additionally, as highlighted in Line 312 of the revised version, we point out that the “airplane” is misclassified as “sky” by the Probability-Level Fusion (PLF) method. **Regarding the number of queries and class probability scores, we clarify that these factors are unrelated.** Specifically, the number of queries defines the maximum number of instances that can be detected, while the number of class probability scores predicted for each query indicates the number of semantic classes the model can recognize. Finally, we explain **why only four segments (sky, grass, and two airplanes in $\bar{M}_i^t$) are presented for 8 queries after Query-Level Fusion (QLF)**. During the fusion process, duplicate segments with high overlap are suppressed, similar to the Non-Maximum Suppression (NMS) used in object detection. We adopt the standard panoptic fusion strategy from Mask2Former for this procedure, and the detailed steps are outlined in **Algorithm 1** in the Appendix of the revised manuscript.

---

> ### Author Response · Authors · 2024-12-01
> **Response to Reviewer fZFv (2/2)**
>
> **Q5:** Ambiguity in Baseline Setting (Table 5)
>
> **A5:** We use the official implementation of CoMFormer as our codebase and treat it as our baseline in **Table 5**. We clarified this in **Line 424** of the revised manuscript.
>
> ---
> **Q6:** Probability Inconsistency Analysis
>
> **A6:** We adopt the *Softmax* activation function to compute the final class scores (added in **Line 269**). In response to your concern, we conduct experiments on the ADE20K *100-10* scenario to compare *Softmax* and *Sigmoid*. We observe that ***Sigmoid* is less effective, whether used with PLF or QLF.** Since *Sigmoid* does not normalize across classes, the class probabilities are independent, which can lead to high confidence values for both base and novel classes, especially when they are semantically similar. For example, in ECLIPSE, the base class “lake” was misclassified as the novel class “water.” As a result, ECLIPSE proposed logit manipulation to alleviate this issue, which **however** brings several hyperparameters $\delta_i (i=0, 1...t)$ at step $t$ that require tailored tuning in different settings, e.g., *100-50*, *100-10*, etc. **In contrast**, *Softmax* can suppress such misclassifications through cross-class normalization. As shown in the table below, our approach with **Softmax (in combination with QLF)** yields significantly better performance without requiring complex hyperparameter tuning.
>
> ||*1-100*|*101-150*|*all*|
> |:--|:--:|:--:|:--:|
> | PLF w/ Sigmoid | 41.3 | 19.5 | 34.0 |
> | QLF w/ Sigmoid | 41.5 | 20.1 | 34.4 |
> | PLF w/ Softmax | **43.1** | 20.8 | 35.7 |
> | **QLF w/ Softmax** | 42.2 | **26.3** | **36.9** |
>
> ---
> ## Additional Comments
> ---
> **Q7:** Experiments in the Disjoint Setting
>
> **A7:** The pioneering work of MiB introduced two distinct settings for continual learning: *disjoint* and *overlap*. Since the *overlap* setting is generally considered more challenging and realistic, we focused primarily on it in our main paper. For completeness, we additionally present experimental results for continual panoptic segmentation on ADE20K under the disjoint setting. The results, as summarized below (**Table 8** in the Appendix of our revised manuscript), highlight the superior performance of ADAPT over existing methods.
>
> |**Method**||**100-5 (11 tasks)** | ||**100-10 (6 tasks)** | || **100-50 (2 tasks)**| |
> |:---------|:-------------------:|:--:|:--:|:-------------------:|:--:|:--:|:-------------------:|:--:|:--:|
> | |*1-100*|*101-150*|*all*|*1-100*|*101-150*|*all*|*1-100*|*101-150*|*all*|
> |MiB|20.5|4.3|15.1|27.7|7.1|20.8|33.7|10.5|26.0|
> |PLOP|19.2|8.8|15.8|28.9|10.6|22.8|34.8|12.4|27.4|
> |CoMFormer|20.1|8.2|16.1|29.7|10.3|23.3|34.7|13.2|27.6|
> |ECLIPSE|34.4|8.9|25.9|34.4|10.2|26.4|35.2|13.3|27.9|
> |**ADAPT (Ours)**|**36.6**|**9.3**|**27.5**|**36.5**|**11.7**|**28.3**|**36.6**|**18.0**|**30.4**|
>
> ---
> **Q8:** Experiments on COCO and robustness on random class ordering
>
> **A8:** Following ECLIPSE, we conduct experiments using two settings (*83-5*, *83-10*) on the **COCO** benchmark. During this process, we identified and fixed two bugs in their implementation: 1) the class indices in COCO start from zero, while their code assumed they start from one, leading to misalignment; and 2) their result summarization for base and novel classes did not account for class order shuffling. After correcting these issues, we obtained the results in the table below (**Table 7** of the revised manuscript), where our approach significantly outperforms other competing methods.
>
> |**Method**| |**83-5(11tasks)**| | |**83-10(6tasks)**| |
> |:--|:--:|:--:|:--:|:--:|:--:|:--:|
> | |*1-83*|*84-133*|*all*|*1-83*|*84-133*|*all*|
> |CoMFormer|33.5|21.9|29.1|37.3|26.6|33.3|
> |ECLIPSE|44.2|18.7|34.6|44.9|21.3|36.0|
> |**ADAPT(Ours)**|**45.3**|**23.2**|**37.0**|**45.7**|**28.8**|**39.3**|
> |Joint|47.1|48.1|47.5|47.1|48.1|47.5|
>
> We examine the **robustness** of our method to different **class ordering** on ADE20K *100-10* scenario. We utilize the 10 randomly shuffled class orders proposed by ECLIPSE for consistency. The PQ distributions are presented in **Figure 3** (see Appendix of the revised manuscript) through boxplots. Notably, our method ADAPT exhibits strong resilience to changes in class ordering, consistently outperforming alternative methods.
>
> ---
> **Q9:** Clarification on Reported Performance Metrics
>
> **A9:** We acknowledge a discrepancy between our reproduced results for ECLIPSE and the metrics reported in their original work. This is because we cannot reproduce their reported results, especially for *100-x* scenarios (2 percent gap). Therefore, for fairness, we report our reproduced results for ECLIPSE. For other methods, we report the official results for consistency. In our revised manuscript, **we have added the official results for ECLIPSE to Table 1**. This inclusion does not alter the demonstrated advantage of ADAPT.

---

> ### Comment · Reviewer_fZFv · 2024-12-02
>
> I appreciate the authors for the comprehensive rebuttals.
> Most concerns are well-addressed, and the quality of the revised paper has improved greatly.
> I raised my rating.
>
> ** Please respond to the reviews from the reviewers c4nN and Bxb3 to solidify your rebuttal as soon as possible. **

---

> > ### Author Response · Authors · 2024-12-03
> > **Thanks for Improving the Rating**
> >
> > We're glad to hear that the revised manuscript has **improved greatly** and that you’ve **increased your rating**.
> >
> > We deeply appreciate your constructive suggestions, which have been invaluable in improving the quality of our manuscript. Thank you for your time and effort in providing such meaningful feedback.
> >
> > Best Regards!

---

### Official Review · Reviewer_Bxb3 · 2024-10-31

**Soundness:** 2
**Presentation:** 2
**Contribution:** 2
**Rating:** 5
**Confidence:** 3

**Summary:**

This paper aims to resolve the efficiency or scalability issues in continual panoptic segmentation methods. A dual-decoder framework is proposed that incorporates attentive self-distillation and prediction fusion to preserve prior knowledge while facilitating model generalization. The majority of pixel decoder's weights are fixed and shared between the teacher and student models. Thus a single forward pass for efficient knowledge distillation can be achieved. Moreover, an attentive self-distillation is introduced to distill useful knowledge from the old classes without distracting from non-object regions. Additionally, a query-level fusion is introduced. to seamlessly integrate the outputs. Experimental evaluation is conducted on ADE20K dataset.

**Strengths:**

1. Clear figure illustration for methods.
2. Interesting idea for probability-level  fusion and query-level fusion.

**Weaknesses:**

Weaknesses:

1. Lack of evidence for Computational Costs and Generalization

2. Need for Experimental Comparison on Computational Overhead

3. Limited Experimental Validation

4. Need for Further Improvement in Table Presentation

These concerns collectively emphasize the need for additional theoretical or experimental evidence and broader validation of the proposed method's claims and generalizability.

**Questions:**

1. [Line 057-058] Is the "raising computational costs" only occur during training? If so, does the computational cost of inference remain same? [Line 060-063] Why continuously introducing additional learnable query features and embeddings for new task is not good? Why this strategy "constrains the model's capacity to generalize to new tasks due to restricted plasticity"? Overall, the reviewer believes that if there are no theoretical or numerical experimental evidence, it would be better to do not judge methods from other areas.
2. [Line 138-139] Since there are many claims that "most of these methods require separate forward passes for the teacher and student models, resulting in considerable computational overhead.", please give clear comparison and experimental results to support.
3. Why the experiments are only conducted on one benchmark? At least two datasets should be involved to validate the generalization ability of the proposed method.
4. It is recommended to adjust the table size and position distribution.

---

> ### Author Response · Authors · 2024-12-02
> **Response to Reviewer Bxb3 (1/2)**
>
> We truly appreciate your positive feedback on the clarity of the figures, as well as on our novel probability-level fusion (PLF) and query-level fusion (QLF) strategy. Your constructive comments have been instrumental in refining our manuscript. Below, we provide detailed responses to the concerns raised:
>
> ---
> ### Weaknesses
> ---
> **Q1:** Lack of evidence for computational costs
>
> **A1:** The table below (**Table 6** in the revised manuscript) highlights the **efficiency** of our approach in both **training** and **inference**. During incremental **training**, ADAPT requires only 2.27 hours for five training steps, representing a **reduction of 34.8\% and 32.4\% in training time** compared to CoMFormer and ECLIPSE, respectively. The reduced training time can be attributed to ADAPT's frozen- and shared-weight design between the teacher and student models, which eliminates the need for dual-model forward passes for knowledge distillation in CoMFormer. Additionally, our efficient adaptation strategy enables a **fourfold reduction in the number of training iterations for each step**, decreasing **from 16k in ECLIPSE to 4k** in our approach. This substantial reduction contributes to the overall decrease in training time.
>
> For **inference**, our method incurs a modest 6.5\% increase in FLOPs compared to ECLIPSE. However, this is accompanied by substantial performance gains, particularly in novel PQ, which improves from **16.5 to 26.3—a remarkable 59.4\% increase**. Similar improvements are observed when compared to CoMFormer. Furthermore, the inference speed of ADAPT remains **comparable** to that of CoMFormer, highlighting the efficiency and practicality of our approach. In contrast, ECLIPSE demonstrates slower inference speeds, likely attributable to its unoptimized implementation. We have added the efficiency analysis to **Sec. 4.5.5**.
>
> |**Method**||**Training** ||| **Inference**|||**100-10 (6 tasks)** ||
> |:-|:-:|:-:|:-:|:-:|:-:|:-:|:-:|:-:|:-:|
> ||#Params|#Iters|Time|FLOPs|GPU Mem.|Time|*1-100*|*101-150*|*all*|
> |CoMFormer|44.38M|**4k**|3.36 hrs|**97.46G**|**4010 MB**|**43.9 ms**|36.0|17.1|29.7|
> |ECLIPSE|**0.55M**|16k|3.48 hrs|99.27G|5407 MB|68.7 ms|41.2|16.5|33.0|
> |**ADAPT (Ours)**|12.12M|**4k**|**2.27 hrs**|105.76G|7127 MB|48.3 ms|**42.2**|**26.3**|**36.9**|
>
> ---
> **Q2:** Limited Experimental Validation (Generalization)
>
> **A2:** Following ECLIPSE, we conduct experiments using two settings (*83-5*, *83-10*) on the **COCO** benchmark. During this process, we identified and fixed two bugs in their implementation: 1) the class indices in COCO start from zero, while their code assumed they start from one, leading to misalignment; and 2) their result summarization for base and novel classes did not account for class order shuffling. After correcting these issues, we obtained the results in the table below (**Table 7** of the revised manuscript), where our approach significantly outperforms other competing methods.
>
> |**Method**| |**83-5(11tasks)**| | |**83-10(6tasks)**| |
> |:--|:--:|:--:|:--:|:--:|:--:|:--:|
> | |*1-83*|*84-133*|*all*|*1-83*|*84-133*|*all*|
> |CoMFormer|33.5|21.9|29.1|37.3|26.6|33.3|
> |ECLIPSE|44.2|18.7|34.6|44.9|21.3|36.0|
> |**ADAPT(Ours)**|**45.3**|**23.2**|**37.0**|**45.7**|**28.8**|**39.3**|
> |Joint|47.1|48.1|47.5|47.1|48.1|47.5|
>
> ---
> **Q3:** Need for Further Improvement in Table Presentation
>
> **A3:** Thank you for your suggestion. We have adjusted the table size to ensure consistent font size across all tables. Additionally, we moved **Table 5** from the original submission to **Table 2** in the revised manuscript so that it is positioned closer to the referencing paragraph in **Sec 4.5.1 Overall Study**. To enhance compactness and improve the overall layout, we have placed **Table 4** and **Table 5** side-by-side horizontally in the revised manuscript.

---

> ### Author Response · Authors · 2024-12-02
> **Response to Reviewer Bxb3 (2/2)**
>
> ---
> ### Questions
> ---
>
> **Q4:** Does the "raising computational costs [Line 057-058]" only occur during training? If so, does the computational cost of inference remain the same?
>
> **A4:** Yes, the dual-model forward pass for the teacher and student models in **CoMFormer** only occurs during training, while the inference cost remains roughly the same with base model Mask2Former. However, we emphasize that the overhead brought by dual-model forward is considerable, leading to a **48\% increase in training time (ADAPT 2.27 hours vs. CoMFormer 3.36 hours)** compared to our frozen- and shared-weight self-distillation mechanism (**ADAPT**), under the ADE20K *100-10* scenario. For a more detailed efficiency comparison, please refer to **A1**.
>
> ---
> **Q5:** Why does continuously introducing additional learnable query features and embeddings for new tasks constrain the model's capacity to generalize to new tasks due to restricted plasticity?
>
> **A5:** ECLIPSE freezes most of the model’s weights and introduces learnable query features and embeddings to adapt to new classes. However, we observe that this approach limits the model's ability to generalize, particularly **when new tasks come with large amounts of training data**. For instance, under ADE20K *100-50* setting, which includes more training data per step compared to *100-10* and *100-5*, **ECLIPSE** performs **poorly** (**23.5 PQ**) on novel classes compared to **CoMFormer** (**27.7 PQ**), which finetunes all model weights. Notably, in Table 1(a) of the ECLIPSE paper, their reported results for CoMFormer are lower than the official results for the ADE20K *100-50* setting. For fairness and consistency, we present the official results in the table below. We observe that the **performance gap on novel PQ** between ECLIPSE and CoMFormer becomes **even more pronounced (21.3 PQ vs. 26.6 PQ for *83-10*) on the COCO benchmark**, which contains significantly more training data (118K images) compared to ADE20K (25K images). These observations suggest that ECLIPSE’s reliance on a limited number of trainable parameters restricts its generalization capacity, especially in tasks with diverse classes and large-scale training data.
>
> |Method| |100-50 (2 tasks)| | |83-5 (11 tasks)| | |83-10 (6 tasks)| |
> |:--|:--:|:--:|:--:|:--:|:--:|:--:|:--:|:--:|:--:|
> | |*1-100*|*101-150*|*all*|*1-83*|*84-133*|*all*|*1-83*|*84-133*|*all*|
> |CoMFormer|41.7 |27.7 |36.7 |33.5|21.9|29.1|37.3|26.6|33.3|
> |ECLIPSE|41.7 | 23.5 |35.6 |44.2|18.7|34.6|44.9|21.3|36.0|
> |**ADAPT(Ours)**|**42.5** |**27.8** | **37.6** |**45.3**|**23.2**|**37.0**|**45.7**|**28.8**|**39.3**|
> |Joint| 43.4 | 32.9 |39.9 |47.1|48.1|47.5|47.1|48.1|47.5|
>
> ---
> **Q6:** Experimental evidence for the claim "most of these methods require separate forward passes for the teacher and student models, resulting in considerable computational overhead."
>
> **A6:** This claim applies to various continual learning methods across domains such as classification, object detection, semantic segmentation, and panoptic segmentation, provided the approach employs a dual-model forward pass during distillation. In the context of panoptic segmentation, we compare our **ADAPT** method to **CoMFormer**, which uses a dual-model forward pass. Since training computational costs include both forward and backward passes, we evaluate efficiency based on the total training time rather than FLOPs, which is typically used for assessing inference costs. In our comparison, with the same number of iterations (4k for each step in the ADE20K *100-10* setting), we find that the dual-model forward pass significantly increases training time, leading to a **48% increase in training time** (ADAPT: 2.27 hours vs. CoMFormer: 3.36 hours). This increase in training time can be attributed to the additional computational cost of managing two model forward passes (teacher and student models). For a more holistic analysis of the efficiency, please refer to **A1**.
>
>
> ---
> **Q7**: At least two datasets should be involved to validate the generalization ability of the proposed method.
>
> **A7**: In addition to **ADE20K**, we have conducted experiments on the **COCO** dataset (see **A2**) to further validate the generalization ability of our proposed method.
>
> ---
> **Q8**: It is recommended to adjust the table size and position distribution.
>
> **A8**: Thank you for your suggestion. Please refer to **A3**.

---

> ### Author Response · Authors · 2024-12-03
> **Reminder: Discussion Phase Ending in 8 Hours**
>
> Dear Reviewer Bxb3,
>
> This is a gentle reminder that the discussion phase will end in **8 hours**. If you have any remaining concerns or questions, please feel free to let us know. We welcome any further discussion and sincerely appreciate your efforts.
>
> Best regards!

---

> ### Author Response · Authors · 2024-12-04
>
> Dear Reviewer Bxb3,
>
> We regret not hearing from you during the author-reviewer discussion period. We understand that this might sometimes happen due to unforeseen circumstances, and we sincerely hope everything is well on your end.
>
> We kindly encourage you to **review our responses to your concerns at your earliest convenience**. We believe our detailed answers address the issues raised and highlight the improvements made to the manuscript. If possible, we would greatly appreciate it if you could **finalize your rating after considering our response**.
>
> Thank you for your time and efforts.
>
> Best regards!

---

### Official Review · Reviewer_c4nN · 2024-11-01

**Soundness:** 3
**Presentation:** 3
**Contribution:** 2
**Rating:** 6
**Confidence:** 3

**Summary:**

The paper proposes a new method for continual panoptic segmentation. This task requires model to be able to adapt to new data and semantic classes, while maintaining the capabilities acquired from previous training stages. A key challenge in this task is catastrophic forgetting, which can result in significant performance degradation, especially for classes from earlier stages that are underrepresented in later stages. The method relies on mask transformer panoptic model and adresses the catastrophic forgetting by careful fine-tuning of only parts of the initial weights. Additionally, it enforces prediction consistency w.r.t. to earlier model instances while emphasizing the loss for informative queries. Finally, the predictions of the initial and the latest model are ensembled to maintain the balance between base and novel classes. The experiments are conducted on ADE20k panoptic dataset following continual learning setups from previous work.

**Strengths:**

The proposed method achieves state-of-the-art performance in continual panoptic segmentation on ADE20k dataset.

The method is computationally more efficient than related work. Due to the freezing of the backbone and the pixel decoder, teachers inference is consisted of only the transformer decoder.

The ablation study reveals positive effects of the proposed contributions on the overall performance.

**Weaknesses:**

Paragraph describing query-level fusion should be better written. Implementation details or some equations might improve clarity of this part.

Presentation of the ablation and validation experiments could be of higher quality. I dont understand the necessity of subsection 4.5. The first table referenced in text is actually table 5, which makes it a bit confusing.

The advantages regarding the computational efficiency are not supported with any experimental results or analysis. Some table comparing training time or FLOPs in a single training iteration with the literature would make this more convincing. Similar analysis would be beneficial for the test-time inference as well. Dual-decoder prediction fusion obviously causes some computational overhead, and this is not measured in any way.

Dual decoder prediction fusion heavily relies on the assumption that most of the data and classes were available in the initial phase. What if this is not the case? Is a setup where most of the data and classes become available in some intermediate step possible in real applications? Perhaps this should be discussed.

The technical novelty of the presented contributions is limited. Self-distillation and weight freezing have already been considered in continual panoptic segmentation.

**Questions:**

Recently, vision-language models such as CLIP caused significant improvements in open-vocabulary panoptic segmentation (e.g. [1])? Would such design solve some technical challenges in continual panoptic segmentation (e.g. adding new classes)? Perhaps this approach trained in a regular way could represent another baseline for CPS?

[1] Yu, Q., He, J., Deng, X., Shen, X., & Chen, L. C. (2023). Convolutions die hard: Open-vocabulary segmentation with single frozen convolutional clip. Advances in Neural Information Processing Systems, 36, 32215-32234.

---

> ### Author Response · Authors · 2024-12-02
> **Response to Reviewer c4nN (1/3)**
>
> We sincerely appreciate your positive feedback on our **impressive performance**, **high efficiency**, and **comprehensive ablation studies**. Your constructive comments have been invaluable in enhancing the quality of our manuscript. Below, we provide detailed responses to the concerns you raised:
>
> ---
> ### Weaknesses
> ---
> **Q1:** Implementation details of query-level fusion (QLF)
>
> **A1:** Our **QLF** is a simple yet effective fusion strategy. As illustrated in Figure 2, we **zero out** the base-class probabilities $p_{i,c}^t (c\in \mathcal{C}^0)$ (represented by the **blue bar** in the bottom-right distribution of Figure 2) predicted by the base decoder $ \mathcal{M}_\text{trans}^t $. This ensures that QLF follows two intuitive rules:
>
> - **The adapted decoder $ \mathcal{M}_\text{trans}^t $ only predicts novel classes**, since base class probabilities are zeroed out.
>
> - **Conversely, $\mathcal{M}_\text{trans}^0$ is solely responsible for predicting the base classes**, as no novel scores are predicted by this decoder.
>
> For illustration, both the base and adapted decoders **each** generate class distribution predictions for 4 queries. The standard panoptic inference fusion strategy from Mask2Former is then applied to combine the predictions from these 8 queries. During fusion, duplicate segments with high overlap are suppressed, similar to the Non-Maximum Suppression (NMS) used in object detection. The detailed fusion procedures are now outlined in **Algorithm 1** in the **Appendix** of the revised manuscript.
>
> ---
> **Q2:** I don‘t understand the necessity of subsection 4.5. The first table referenced in the text is actually Table 5, which makes it a bit confusing.
>
> **A2:** Thank you for bringing this to our attention. We apologize for the layout mistake. To clarify, subsection 4.5 "Ablation Study" actually consists of several **subsubsections**, namely 4.5.1 "Overall Study," 4.5.2 "Adaptation Strategy," 4.5.3 "Attentive Self-Distillation," 4.5.4 "Dual-Decoder Prediction Fusion," and 4.5.5 "Efficiency," which were **originally mistakenly treated as separate subsections**.
>
> Additionally, we moved **Table 5** from the original submission to **Table 2** in the revised manuscript so that it is positioned closer to the referencing paragraph in **Sec 4.5.1 Overall Study**. To enhance compactness and improve the overall layout, we have placed **Table 4** and **Table 5** side-by-side horizontally in the revised manuscript. Finally, we have adjusted the table size to ensure consistent font size across all tables.
>
> ---
> **Q3:** Training and inference efficiency analysis
>
> **A3:** The table below (**Table 6** in the revised manuscript) highlights the **efficiency** of our approach in both **training** and **inference**. During incremental **training**, ADAPT requires only 2.27 hours for five training steps, representing a **reduction of 34.8\% and 32.4\% in training time** compared to CoMFormer and ECLIPSE, respectively. The reduced training time can be attributed to ADAPT's frozen- and shared-weight design between the teacher and student models, which eliminates the need for dual-model forward passes for knowledge distillation in CoMFormer. Additionally, our efficient adaptation strategy enables a **fourfold reduction in the number of training iterations for each step**, decreasing **from 16k in ECLIPSE to 4k** in our approach. This substantial reduction contributes to the overall decrease in training time.
>
> For **inference**, our method incurs a modest 6.5\% increase in FLOPs compared to ECLIPSE. However, this is accompanied by substantial performance gains, particularly in novel PQ, which improves from **16.5 to 26.3—a remarkable 59.4\% increase**. Similar improvements are observed when compared to CoMFormer. Furthermore, the inference speed of ADAPT remains **comparable** to that of CoMFormer, highlighting the efficiency and practicality of our approach. In contrast, ECLIPSE demonstrates slower inference speeds, likely attributable to its unoptimized implementation. We have added the efficiency analysis to **Sec. 4.5.5**.
>
> |**Method**||**Training** ||| **Inference**|||**100-10 (6 tasks)** ||
> |:-|:-:|:-:|:-:|:-:|:-:|:-:|:-:|:-:|:-:|
> ||#Params|#Iters|Time|FLOPs|GPU Mem.|Time|*1-100*|*101-150*|*all*|
> |CoMFormer|44.38M|**4k**|3.36 hrs|**97.46G**|**4010 MB**|**43.9 ms**|36.0|17.1|29.7|
> |ECLIPSE|**0.55M**|16k|3.48 hrs|99.27G|5407 MB|68.7 ms|41.2|16.5|33.0|
> |**ADAPT (Ours)**|12.12M|**4k**|**2.27 hrs**|105.76G|7127 MB|48.3 ms|**42.2**|**26.3**|**36.9**|

---

> ### Author Response · Authors · 2024-12-02
> **Response to Reviewer c4nN (2/3)**
>
> **Q4:** Is the setup where most of the data and classes become available in some intermediate step possible in real applications?
>
> **A4:** In real-world applications, it is **uncommon** for most data and classes to become available only at an intermediate stage. Typically, models are deployed after being trained on a sufficient scale of data to ensure **robustness**, and new data or classes are introduced gradually over time. However, if a situation does arise where a substantial portion of data and classes is introduced only at a later stage, it may be more effective to retrain the model from scratch using all available data. This is because early-stage training on **limited data** might **NOT** provide **robust knowledge** for transferring to new tasks, potentially leading to suboptimal performance. Moreover, since the early-stage training does NOT involve large-scale data, storing these old data and retraining a new model from scratch along with new data is **affordable** and **cost-efficient**.
>
> ---
> **Q5:** The technical novelty of the presented contributions is limited. Self-distillation and weight freezing have already been considered in continual panoptic segmentation.
>
> **A5:** We respectfully disagree with the claim that the novelty of our contributions is limited. While self-distillation and weight freezing have been explored in prior works like CoMFormer and ECLIPSE, our approach introduces key advancements in both **methodology** and **efficiency**.
>
> First, **weight freezing** in continual panoptic segmentation is non-trivial. In ECLIPSE, freezing most of the model weights limits generalization, especially when new tasks involve large amounts of training data (**see Reviewer Bxb3 Q5**). Conversely, updating all parameters, as in CoMFormer, risks poor knowledge retention due to catastrophic forgetting. Our comprehensive ablation studies in **Table 3** show that our freezing strategy (only fine-tuning cross-attention layers and FFNs) strikes an effective balance between **plasticity** (learning new tasks) and **rigidity** (retaining old knowledge). Additionally, we provide a **theoretical hypothesis** for the effectiveness of our adaptation strategy (see **Reviewer fZFv Q1**).
>
> Regarding **self-distillation**, while it has been employed in CoMFormer, CoMFormer suffers from significant computational overhead due to the dual-model forward pass during knowledge distillation (KD). We address this by introducing a **frozen- and shared-weight strategy**, eliminating the need for dual-model forward passes and **reducing training time by 34.8%** on ADE20K *100-10* (as detailed in **Q3**). Unlike CoMFormer, which applies **unbiased KD** [1] to all queries, our approach selectively applies our **attentive KD** to queries that are not matched with any new classes. Notably, our attentive KD incorporates a **modulating term $\alpha(1-\hat{p}_x^{t-1} (i,\emptyset))^\gamma$** to **emphasize informative queries and down-weight less useful ones** based on background confidence (see **Sec. 3.3** for details). These two mechanisms facilitate more efficient and effective retention of old knowledge, alleviating the natural bias toward current novel classes.
>
> Finally, our method **ADAPT** consistently achieves **state-of-the-art** results on **ADE20K** and **COCO** benchmarks across various settings with **high efficiency**. We believe these contributions (**acknowledged by Reviewer fZFv with confidence 5**) are significant and provide a meaningful improvement over existing methods [2-3] in terms of both **performance** and **efficiency**.
>
> ---
> [1] Cermelli, Fabio, et al. "Modeling the background for incremental learning in semantic segmentation." Proceedings of the IEEE/CVF Conference on Computer Vision and Pattern Recognition. 2020.
>
> [2] Cermelli, Fabio, Matthieu Cord, and Arthur Douillard. "Comformer: Continual learning in semantic and panoptic segmentation." Proceedings of the IEEE/CVF Conference on Computer Vision and Pattern Recognition. 2023.
>
> [3] Kim, Beomyoung, Joonsang Yu, and Sung Ju Hwang. "ECLIPSE: Efficient Continual Learning in Panoptic Segmentation with Visual Prompt Tuning." Proceedings of the IEEE/CVF Conference on Computer Vision and Pattern Recognition. 2024.

---

> ### Author Response · Authors · 2024-12-02
> **Response to Reviewer c4nN (3/3)**
>
> ---
> ### Questions
> ---
> **Q6:** Would the open-vocabulary panoptic segmentation method [4] solve some technical challenges in continual panoptic segmentation (e.g. adding new classes)? Perhaps this approach trained in a regular way could represent another baseline for CPS?
>
> **A6:** We thank the reviewer for bringing this approach [4] for open-vocabulary panoptic segmentation to our attention. [4] introduces an out-vocab branch, allowing the model to handle open vocabularies. However, we believe this approach is NOT suitable as a baseline for continual panoptic segmentation (CPS).
>
> If we train the in-vocab branch using novel data at each step in a conventional manner, this setup becomes functionally identical to the Mask2Former framework, which is essentially fine-tuning (**FT**) in **Table 1**. On the other hand, if we utilize the out-vocab branch for the **geometric ensemble** (Figure 3 in [4]) to handle novel classes, the approach relies on **additional information** from the pretrained CLIP text encoder. This external knowledge makes a direct comparison **unfair**, as other CPS methods [2-3], including ours, do NOT have access to such extra information.
>
> Nevertheless, we acknowledge that open-vocabulary learning is an interesting topic, particularly in the context of large foundation models. Given this, we will cite [4] and discuss it in the related work section of our final manuscript for completeness.
>
> ---
> [2] Cermelli, Fabio, Matthieu Cord, and Arthur Douillard. "Comformer: Continual learning in semantic and panoptic segmentation." Proceedings of the IEEE/CVF Conference on Computer Vision and Pattern Recognition. 2023.
>
> [3] Kim, Beomyoung, Joonsang Yu, and Sung Ju Hwang. "ECLIPSE: Efficient Continual Learning in Panoptic Segmentation with Visual Prompt Tuning." Proceedings of the IEEE/CVF Conference on Computer Vision and Pattern Recognition. 2024.
>
> [4] Yu, Q., He, J., Deng, X., Shen, X., & Chen, L. C. (2023). Convolutions die hard: Open-vocabulary segmentation with single frozen convolutional clip. Advances in Neural Information Processing Systems, 36, 32215-32234.

---

> > ### Comment · Reviewer_c4nN · 2024-12-02
> >
> > I would like to thank the authors for their effort during the rebuttal. The revised version of the paper clarifies some implementation details and now experimentally supports the claims regarding the efficiency. Therefore, I improved my score.

---

> > > ### Author Response · Authors · 2024-12-03
> > > **Thanks for Improving your Rating**
> > >
> > > Thank you for your positive feedback and for **raising your score**. We are grateful for your time and effort in reviewing our manuscript. We're glad to hear that the revisions have **clarified the implementation details** and that the **experimental results** now effectively **support our claims regarding efficiency**.
> > >
> > > Your constructive comments have played a crucial role in improving the quality of our work.
> > >
> > > Best Regards!

---

> ### Author Response · Authors · 2024-12-04
>
> Hi Reviewer c4nN,
>
> Could you kindly re-evaluate the **confidence** and **contribution** scores? We believe our responses have addressed your concerns and improved your confidence in the contribution of our work. Thank you.
>
> Best Regards,
>
> The Authors

---

### Official Review · Reviewer_diCs · 2024-11-09

**Soundness:** 3
**Presentation:** 3
**Contribution:** 3
**Rating:** 6
**Confidence:** 4

**Summary:**

This paper addresses the challenges in continual panoptic segmentation (CPS). In order to tackle catastrophic forgetting, the authors propose a dual-decoder Mask2Former framework combined with attentive self-distillation (called ADAPT) to efficiently retain past knowledge and generalize to new classes.

**Strengths:**

- The reviewer found the paper to be well-written and easy to understand.

- The reviewer likes the use of a dual-decoder framework that allows for knowledge retention with minimal computational costs.

- The paper also presents a solid self-distillation mechanism focuses on informative regions by down-weighting non-object areas, making knowledge retention more targeted. The reviewer finds the query-level fusion (QLF) strategy to eliminate the scale mismatch issues commonly found in probability-level fusion to be interesting.

**Weaknesses:**

- Although the reviewer likes the idea of freezing the encoder as it reduces computational load and helps retain base knowledge, it may restrict the model's ability to adapt fully to new classes over very long sequences of tasks, particularly as the diversity or complexity of new classes increases. The authors can comment on this.

- On a similar note, while the authors report results with ResNet-50 backbone, it’s unclear to the reviewer how the method scales with larger models or higher-resolution images. The freezing and distillation mechanisms might still impose computational and memory limitations in more demanding setups.

- The author's might benefit from observing the effect of their method on open-vocab evaluation settings. Further, it may be interesting to evaluate on standard settings as well. For example, on COCO dataset with the number of classes growing in the number of tasks.

**Questions:**

N/A

---

> ### Author Response · Authors · 2024-12-01
> **Response to Reviewer diCs (1/2)**
>
> We sincerely appreciate your positive feedback on our dual-decoder design, self-distillation mechanism, and query-level fusion (QLF) strategy. Your constructive comments have been invaluable in refining our manuscript. Below, we provide detailed responses to address the concerns raised:
>
> ---
> **Q1:** Freezing the encoder may limit the model's ability to adapt fully to new classes over long task sequences, especially as the diversity or complexity of new classes increases.
>
> **A1:** We compare the performance of **freezing the encoder (ADAPT)** versus **leaving it unfrozen (Unfrozen encoder)** in long-sequence tasks, specifically on the *100-5* (11 tasks) setting on **ADE20K** and *83-5* (11 tasks) on **COCO**, where COCO involves more complex scenes. The results below indicate that freezing the encoder does not hinder the model’s generalization capacity. In fact, the **transformer decoder itself remains highly adaptable to new tasks**. In contrast, unfreezing the encoder results in catastrophic forgetting and renders our shared self-distillation mechanism not applicable. As a consequence, a dual-model forward pass (teacher and student models) becomes necessary for distillation, similar to CoMFormer. This incurs additional computational costs during training.
>
> |**Method**| |**100-5 (11 tasks)**| | |**83-5 (11 tasks)**| |
> |:--|:--:|:--:|:--:|:--:|:--:|:--:|
> | |*1-100*|*101-150*|*all*|*1-83*|*84-133*|*all*|
> |Unfrozen encoder |  28.8 |  21.7 | 26.4 | 30.1 | 17.9 | 25.5 |
> |**ADAPT (Ours)** | **42.2** | **26.3** | **36.9** |**45.3**|**23.2**|**37.0**|
>
> ---
> **Q2:** The freezing and distillation mechanisms might still impose computational and memory limitations in more demanding setups. How does the method scale with larger models or higher-resolution images?
>
> **A2:** The table below (**Table 6** in the revised manuscript) highlights the **efficiency** of our approach in both **training** and **inference**. During **training**, we first clarify that the freezing mechanism actually reduces computational costs, as no gradients are computed for the frozen weights during backpropagation. Compared to CoMFormer, our distillation mechanism shares the computation up to the transformer decoder, resulting in **1.09 hours (34.8%) less training time**. While our per-iteration FLOPs are slightly higher than ECLIPSE due to adaptable self-attention layers and FFNs, our **efficient adaptation strategy** enables a **fourfold reduction in training iterations per step** (from 16k in ECLIPSE to 4k in our approach). This results in an **overall reduction in training time (ADAPT: 2.27 hours vs. ECLIPSE 3.48 hours).**
>
> For **inference**, our method incurs a modest 6.5\% increase in FLOPs compared to ECLIPSE. However, this is accompanied by substantial performance gains, particularly in novel PQ, which improves from **16.5 to 26.3—a remarkable 59.4\% increase**. Similar improvements are observed when compared to CoMFormer. Furthermore, the inference speed of ADAPT remains **comparable** to that of CoMFormer, highlighting the efficiency and practicality of our approach. In contrast, ECLIPSE demonstrates slower inference speeds, likely attributable to its unoptimized implementation. We have added the efficiency analysis to **Sec. 4.5.5**.
>
> |**Method**||**Training** ||| **Inference**|||**100-10 (6 tasks)** ||
> |:-|:-:|:-:|:-:|:-:|:-:|:-:|:-:|:-:|:-:|
> ||#Params|#Iters|Time|FLOPs|GPU Mem.|Time|*1-100*|*101-150*|*all*|
> |CoMFormer|44.38M|**4k**|3.36 hrs|**97.46G**|**4010 MB**|**43.9 ms**|36.0|17.1|29.7|
> |ECLIPSE|**0.55M**|16k|3.48 hrs|99.27G|5407 MB|68.7 ms|41.2|16.5|33.0|
> |**ADAPT (Ours)**|12.12M|**4k**|**2.27 hrs**|105.76G|7127 MB|48.3 ms|**42.2**|**26.3**|**36.9**|
>
> **Scalability:** We note that the **scaling behavior** for CoMFormer, ECLIPSE, and ADAPT is **consistent** when using larger backbone models or higher-resolution images. This is because all three methods share the same architecture of the encoder and pixel decoder, which can encode the input images of various resolutions into feature maps of consistent resolution. Therefore, it’s feasible to measure the efficiency in the conventional setting as in prior works CoMFormer and ECLIPSE to ensure consistency.

---

> ### Author Response · Authors · 2024-12-01
> **Response to Reviewer diCs (2/2)**
>
> **Q3:** Evaluation on the COCO dataset and the open-vocab setting.
>
> **A3:** Following ECLIPSE, we conduct experiments using two settings (*83-5*, *83-10*) on the **COCO** benchmark. During this process, we identified and fixed two bugs in their implementation: 1) the class indices in COCO start from zero, while their code assumed they start from one, leading to misalignment; and 2) their result summarization for base and novel classes did not account for class order shuffling. After correcting these issues, we obtained the results in the table below (**Table 7** of the revised manuscript), where our approach significantly outperforms other competing methods.
>
> |**Method**| |**83-5(11tasks)**| | |**83-10(6tasks)**| |
> |:--|:--:|:--:|:--:|:--:|:--:|:--:|
> | |*1-83*|*84-133*|*all*|*1-83*|*84-133*|*all*|
> |CoMFormer|33.5|21.9|29.1|37.3|26.6|33.3|
> |ECLIPSE|44.2|18.7|34.6|44.9|21.3|36.0|
> |**ADAPT(Ours)**|**45.3**|**23.2**|**37.0**|**45.7**|**28.8**|**39.3**|
> |Joint|47.1|48.1|47.5|47.1|48.1|47.5|
>
> Open-vocabulary models are designed to generalize to unseen classes during inference, typically using *text prompts* or *semantic embeddings*. In contrast, continual learning models focus on retaining knowledge from previously seen tasks while adapting to new ones, which **do NOT gain the ability to recognize open classes without adaptation**. Given the distinct objectives of these two types of models, it is not appropriate to directly compare them. **This probably explains why prior works CoMFormer and ECLIPSE did NOT validate under open-vocabulary settings.** Nevertheless, we acknowledge that open-vocabulary learning is an important topic, particularly in the context of large foundation models. We are interested in exploring and investigating open-vocabulary learning in our future research.

---

> > ### Comment · Reviewer_diCs · 2024-12-01
> > **Response to Authors**
> >
> > Hi Authors,
> >
> > Thank you for your responses and your hard work. I have increased my confidence rating from 3 to 4 (would have increased the score as well if there was 1 point jump allowed beyond rating 6). All the best!

---

> ### Author Response · Authors · 2024-12-02
> **Thanks for the confidence improvement from 3 to 4 and the intended score raising to 7**
>
> We’re glad to hear that our responses have addressed your concerns, and we truly appreciate your **increased confidence** in the paper. We are grateful for your **improved recognition**, even though the rating system does NOT allow for a **1-point jump to 7**.
>
> We sincerely appreciate your constructive feedback and thoughtful suggestions, which have helped improve our work.
>
> Best wishes.

---

### Meta-Review · Area_Chair_pErB · 2024-12-24

**Metareview:**

The paper proposes a novel approach for continual panoptic segmentation using a dual-decoder framework with attentive self-distillation and prediction fusion to address catastrophic forgetting and scalability issues. The method emphasizes efficient knowledge distillation and computational stability while achieving strong empirical results on the ADE20K benchmark. It combines innovative mechanisms like query-level fusion and weight freezing to balance knowledge retention and adaptability.

Reviewer Bxb3 maintains a position to reject the paper due to concerns about limited evidence supporting computational efficiency claims, insufficient generalization validation, and the reliance on a single dataset (ADE20K) for evaluation. Additional criticisms include a lack of comparative analysis on computational overhead and inadequate table presentation. While the authors provided comprehensive responses, including expanded experiments on COCO, detailed efficiency analyses, and improved presentation, Bxb3 did not participate in the discussion phase, leaving their concerns unresolved. The authors addressed most of Bxb3’s criticisms effectively. They demonstrated significant computational savings during training and inference and included results from multiple benchmarks to validate generalization. However, due to the lack of discussion feedback from Bxb3, it remains unclear whether their concerns about broader applicability and methodological rigor were fully satisfied.

Considering the comprehensive rebuttal and the additional evidence provided, the paper presents meaningful contributions and demonstrates practical relevance in continual learning. While unresolved issues with one reviewer persist, the strong support from other reviewers and the addressed concerns justify acceptance.

**Additional Comments On Reviewer Discussion:**

Please refer to the meta-review.

---

### Decision · Program_Chairs · 2025-01-22

Accept (Poster)